# Universal Joint Approximation of Manifolds and Densities by Simple Injective Flows

## Abstract

We analyze neural networks composed of bijective flows and injective expansive elements. We find that such networks universally approximate a large class of manifolds simultaneously with densities supported on them. Among others, our results apply to the well-known coupling and autoregressive flows. We build on the work of Teshima et al. 2020 on bijective flows and study injective architectures proposed in Brehmer et al. 2020 and Kothari et al. 2021. Our results leverage a new theoretical device called the embedding gap, which measures how far one continuous manifold is from embedding another. We relate the embedding gap to a relaxation of universally we call the manifold embedding property, capturing the geometric part of universality. Our proof also establishes that optimality of a network can be established "in reverse," resolving a conjecture made in Brehmer et al. 2020 and opening the door for simple layer-wise training schemes. Finally, we show that the studied networks admit an exact layer-wise projection result, Bayesian uncertainty quantification, and black-box recovery of network weights.

## 1 Introduction

In the past several years, invertible flow networks emerged as powerful deep learning models to learn maps between distributions (Durkan et al., 2019a; Grathwohl et al., 2018; Huang et al., 2018; Jaini et al., 2019; Kingma et al., 2016; Kingma & Dhariwal, 2018; Kobyzev et al., 2020; Kruse et al., 2019; Papamakarios et al., 2019). They generate excellent samples (Kingma & Dhariwal, 2018) and facilitate solving scientific inference problems (Brehmer & Cranmer, 2020; Kruse et al., 2021).

By design, invertible flows are bijective and, hence, may not be a natural choice when the target distribution has low-dimensional support. This problem can be overcome by combining bijective flows with expansive, injective layers, which map to higher dimensions (Brehmer & Cranmer, 2020; Cunningham et al., 2020; Kothari et al., 2021). Despite their empirical success, the theoretical aspects of such globally injective architectures are not well understood.

In this work, we present approximation-theoretic properties of injective flows whose architecture combines bijective flows and expansive, injective layers, with an emphasis on approximating measures with low-dimensional support. We state conditions under which these networks are universal approximators and describe how their design enables applications to inference and inverse problems.

### 1.1 Prior Work

The idea to combine invertible (coupling) layers with expansive layers has been explored by Brehmer & Cranmer (2020) and Kothari et al. (2021). Brehmer & Cranmer (2020) combine two flow networks with a simple expansive element (in the sense made precise in Section 2.1) and obtain a network that paramterizes probability distributions supported on manifolds. They suggest that such constructions may be universal but neither they not Kothari et al. (2021) derive theoretical results. We discuss in detail the connection between these two empirical works and the approximation results derived here in Appendix B.1.

Kothari et al. (2021) propose expansive coupling layers and build networks similar to that of Brehmer & Cranmer (2020) but with an arbitrary number of expressive and expansive elements. They observe that the resulting network trains much faster with a smaller memory footprint, while producing high-quality samples on a variety of benchmark datasets.

While to the best of our knowledge, there are no approximation-theoretic results for injective flows, there exists a body of work on universality of invertible flows; see Kobyzev et al. (2020) for an overview. Several works show that certain bijective flow architectures are distributionally universal. This was proved for autoregressive flows with sigmoidal activations by Huang et al. (2018) and for sum-of-squares polynomial flows by Jaini et al. (2019). Teshima et al. (2020) show that several flow networks including those from Huang et al. (2018) and Jaini et al. (2019) are also universal approximators of diffeomorphisms.

The injective flows considered here have key applications in inference and inverse problems; for an overview of deep learning approaches to inverse problems, see Arridge et al. (2019). Bora et al. (2017) proposed to regularize compressed sensing problems by constraining the recovery to the range of (pre-trained) generative models. Injective flows with efficient inverses as generative models gives an algorithmic projection[1] on the range, which facilitates implementation of reconstruction algorithms. An alternative approach is Bayesian, where flows are used to obtain tractable variational approximations of posterior distributions over parameters of interest, via supervised training on labeled input-output data pairs. Ardizzone et al. (2018) encode the dimension-reducing forward process by an invertible neural network (INN), with additional outputs used to encode posterior variability. Invertibility guarantees that a model of the inverse process is learned implicitly. For a given measurement, the inverse pass of the INN approximates the posterior over parameters. Sun & Bouman (2020) propose variational approximations of the posterior using an untrained deep generative model. They train a normalizing flow which produces samples from the posterior, with the prior and the noise model given implicitly by the regularized misfit functional. In Kothari et al. (2021) this procedure is adapted to priors specified by injective flows which yields significant improvements in computational efficiency.

## 1.2 OUR CONTRIBUTION

We derive new approximation results for neural networks composed of bijective flows and injective expansive layers, including those introduced by Brehmer & Cranmer (2020) and Kothari et al. (2021). We show that these networks universally jointly approximate a large class of manifolds and densities supported on them.

We build on the results of Teshima et al. (2020) and develop a new theoretical device which we refer to as the embedding gap. This gap is a measure of how nearly a mapping from $\mathbb{R}^o \to \mathbb{R}^m$ embeds an $n$-dimensional manifold in $\mathbb{R}^m$, where $n \leq o$. We find a natural relationship between the embedding gap and the problem of approximating probability measures with low-dimensional support.

We then relate the embedding gap to a relaxation of universality we call *manifold embedding property*. We show that this property captures the essential geometric aspects of universality and uncover important topological restrictions on the approximation power of these networks, to our knowledge, heretofore unknown in the literature. We give an example of an absolutely continuous measure $\mu$ and embedding $f \colon \mathbb{R}^2 \to \mathbb{R}^3$ such that $f_\# \mu$ can not be approximated with combinations of flow layers and linear expansive layers. This may be surprising since it was previously thought that networks such as those of Brehmer & Cranmer (2020) can approximate any "nice" density supported on a "nice" manifold. We establish universality for manifolds with suitable topology, described in terms of *extendable embeddings*. Our proof shows that optimality of the approximating network can be established in reverse: optimality of a given layer can be established without optimality of preceding layers. This settles a (generalization of a) conjecture posed for a two-layer case in (Brehmer & Cranmer, 2020). Finally, we show that these universal architectures are also practical and admit exact layer-wise projections, as well as other properties discussed in Section 3.4.

## 2 DESCRIPTION OF THE ARCHITECTURE

Let $C(X, Y)$ denote the space of continuous functions $X \to Y$. We study networks in $\mathcal{F} \subset C(X, Y)$ that are of the form:

$$\mathcal{F} = \mathcal{T}_L^{n_L} \circ \mathcal{R}_L^{n_{L-1}, n_L} \circ \cdots \circ \mathcal{T}_1^{n_1} \circ \mathcal{R}_1^{n_0, n_1} \circ \mathcal{T}_0^{n_0} \tag{1}$$

---

[1]Idempotent but in general not orthogonal.

where $\mathcal{R}_\ell^{n_{\ell-1}, n_\ell} \subset C(\mathbb{R}^{n_{\ell-1}}, \mathbb{R}^{n_\ell})$, $\mathcal{T}_\ell^{n_\ell} \subset C(\mathbb{R}^{n_\ell}, \mathbb{R}^{n_\ell})$, $L \in \mathbb{N}$ is the number of networks, $n_0 = n$, $n_L = m$, and $n_\ell \geq n_{\ell-1}$ for $\ell = 1, \ldots, L$. We introduce a well-tuned shorthand notation and write $\mathcal{H} \circ \mathcal{G} := \{h \circ g \colon h \in \mathcal{H}, g \in \mathcal{G}\}$ throughout the paper.

We identify $\mathcal{R}$ with the expansive layers and $\mathcal{T}$ with the bijective flows. Loosely speaking, the purpose of the expansive layers is to allow the network to parameterize high-dimensional functions by low-dimensional coordinates in an injective way. The flow networks give the network the expressivity necessary for universal approximation of manifold-supported distributions.

## 2.1 EXPANSIVE LAYERS

The expansive elements transform an $n$-dimensional manifold $\mathcal{M}$ embedded in $\mathbb{R}^{n_{\ell-1}}$, and embed it in a higher dimensional space $\mathbb{R}^{n_\ell}$. To preserve the topology of the manifold, this must be done injectively. We thus make the following assumptions about the expansive elements:

**Definition 1** (Expansive Element). Let $\ell = 1, \ldots, L$, and $\mathcal{R}_\ell^{n_{\ell-1}, n_\ell}$ be a family of functions from $\mathbb{R}^{n_{\ell-1}} \to \mathbb{R}^{n_\ell}$. $\mathcal{R}_\ell^{n_{\ell-1}, n_\ell}$ is a family of expansive elements if every $R \in \mathcal{R}_\ell^{n_{\ell-1}, n_\ell}$ is injective and Lipschitz.

Examples of expansive elements include

(R1) Zero padding: $R(x) = \left[x^T, \mathbf{0}^{(m-n)}\right]^T$ where $\mathbf{0}^{(m-n)}$ is the zero vector (Brehmer & Cranmer, 2020).

(R2) Multiplication by an arbitrary full-rank matrix, or one-by-one convolution:

$$R(x) = Wx, \quad \text{or} \quad R(x) = w \star x \tag{2}$$

where $W \in \mathbb{R}^{m \times n}$ and $\operatorname{rank}(W) = n$ (Cunningham et al., 2020), and $w$ is a convolution kernel $\star$ denotes convolution Kingma & Dhariwal (2018).

(R3) Injective ReLU layers: $R = \operatorname{ReLU}(Wx)$, $W = \left[B^T, -DB^T, M^T\right]^T$, $R(x) = \operatorname{ReLU}\left(\left[w^T, -w^T\right] \star x\right)$ for matrix $B \in \operatorname{GL}_n(\mathbb{R})$, positive diagonal matrix $D \in \mathbb{R}^{n \times n}$, and arbitrary matrix $M \in \mathbb{R}^{(m-2n) \times n}$ (Puthawala et al., 2020).

(R4) Injective ReLU networks (Puthawala et al., 2020, Theorem 15). These are functions $R \colon \mathbb{R}^n \to \mathbb{R}^m$ of the form $R(x) = W_{L+1} \operatorname{ReLU}(\ldots \operatorname{ReLU}(W_1 x + b_1) \ldots) + b_L$ where $W_\ell$ are $n_{\ell+1} \times n_\ell$ matrices and $b_\ell$ are the bias vectors in $\mathbb{R}^{n_{\ell+1}}$. The weight matrices $W_L$ satisfy the Directed Spanning Set (DSS) condition for $\ell \leq L$ (that make all layers injective) and $W_{L+1}$ is a generic matrix which makes the map $R \colon \mathbb{R}^n \to \mathbb{R}^m$ injective. Note that the DSS condition requires that $n_\ell \geq 2n_{\ell-1} + 1$ for $\ell \leq L$ and we have $n_1 = n$ and $n_{L+1} = m$.

## 2.2 BIJECTIVE FLOW NETWORKS

The bulk of our theoretical analysis is devoted to expressive elements. The expressive elements bend the range of the expansive elements into the correct shape. We make the following assumptions about the expressive elements:

**Definition 2** (Bijective Flow Network). Let $\ell = 0, \ldots, L$ be given and let $\mathcal{T}_\ell^{n_\ell} \subset C(\mathbb{R}^{n_\ell}, \mathbb{R}^{n_\ell})$. $\mathcal{T}_\ell^{n_\ell}$ is a family of bijective flow networks if every $T \in \mathcal{T}_\ell^{n_\ell}$ is Lipschitz continuous and bijective.

Examples of bijective flow networks include

(T1) *Coupling flows*, introduced by Dinh et al. (2014) consider $R(\mathbf{x}) = H_k \circ \cdots \circ H_1(\mathbf{x})$ where

$$H_i(\mathbf{x}) = \begin{bmatrix} h_i\left([\mathbf{x}]_{1:d}, g_i\left([\mathbf{x}]_{d+1:n}\right)\right) \\ [\mathbf{x}]_{d+1:n} \end{bmatrix}. \tag{3}$$

In Eqn. 3, $h_i \colon \mathbb{R}^d \times \mathbb{R}^e \to \mathbb{R}^d$ is invertible w.r.t. the first argument given the second, and $g_i \colon \mathbb{R}^{n-d} \to \mathbb{R}^e$ is arbitrary. Typically in practice the operation in Eqn. 3 is combined with additional invertible operations such as permutations, masking or convolutions Dinh et al. (2014; 2016); Kingma & Dhariwal (2018).

(T2) *Autoregressive flows*, introduced by Kingma et al. (2016) are generalizations of triangular flows $A \colon \mathbb{R}^n \to \mathbb{R}^n$ where for $i = 1, \ldots, n$ the $i$'th value of $A$ is given by of the form

$$[A]_i(\mathbf{x}) = h_i\left([\mathbf{x}]_i, g_i\left([\mathbf{x}]_{1:i-1}\right)\right) \tag{4}$$

In Eqn. 4, $h_i \colon \mathbb{R} \times \mathbb{R}^m \to \mathbb{R}$ where again $h_i$ is invertible w.r.t. the first argument given the second, and $g_i \colon \mathbb{R}^{i-1} \to \mathbb{R}^m$ is arbitrary except for $g_1 = \mathbf{0}$. In Huang et al. (2018), the authors choose $h_i(x, \mathbf{y})$, where $\mathbf{y} \in \mathbb{R}^m$, to be a multi-layer perceptron (MLP) of the form

$$h_i(x, \mathbf{y}) = \phi \circ W_{p,\mathbf{y}} \circ \cdots \circ \phi \circ W_{1,\mathbf{y}}(x) \tag{5}$$

where $\phi$ is a sigmoidal increasing non-linear activation function.

# 3 MAIN RESULTS

## 3.1 EMBEDDING GAP

We call a function $f$ an embedding and denote it by $f \in \mathrm{emb}(X, Y)$ if $f : X \to Y$ is continuous, injective, and $f^{-1} \colon f(X) \to X$ is continuous[2]. Also we denote $\mathrm{emb}^k(\mathbb{R}^n, \mathbb{R}^m) = \mathrm{emb}(\mathbb{R}^n, \mathbb{R}^m) \cap C^k(\mathbb{R}^n, \mathbb{R}^m)$. In order to set up our result concerning embedding of manifolds, we first need a way to measure the degree to which a mapping $g \in \mathrm{emb}(\mathbb{R}^o, \mathbb{R}^m)$ nearly embeds a manifold $\mathcal{M} = f(K)$ for compact $K \subset \mathbb{R}^n$ and $f \in \mathrm{emb}(K, \mathbb{R}^m)$. With this in mind we introduce *embedding gap* $B_{K,W}(f, g)$, a non-symmetric notion of distance between $f$ and $g$. Later in the paper, $f$ will be the function to be approximated, and $g$ a flow-network to be the approximator.

**Definition 3** (Embedding Gap). Let $K \subset \mathbb{R}^n$ be compact and non-empty, $W \subset \mathbb{R}^o$ contain the closure of set $U$ which is open in the subspace topology of some vector subspace $V$ of dimension $p$, where $n \leq p \leq o \leq m$, $f \in \mathrm{emb}(K, \mathbb{R}^m)$, and $g \in \mathrm{emb}(W, \mathbb{R}^m)$. Then we define the Embedding Gap between $f$ and $g$ on sets $K$ and $W$ as

$$B_{K,W}(f, g) = \inf_{r \in \mathrm{emb}(f(K), g(W))} \|I - r\|_{L^\infty(f(K))} \tag{6}$$

where $I \colon f(K) \to f(K)$ is the identity function and $\|h\|_{L^\infty(X)} = \mathrm{ess\,sup}_{x \in X} \|h(x)\|_2$ for $h \colon X \to Y$. We refer to the embedding gap between $f$ and $g$ without specifying $K$ and $W$ when it is clear from context.

*Remark* 1. As $W \subset \mathbb{R}^o$ contains $U$, an open set in $V$, there is an affine map $A : \mathbb{R}^n \to V$ such that $A(K) \subset W$. Then, the map $r_0 = g \circ A \circ f^{-1} : f(K) \to g(W)$ is an injective continuous map from a compact set to its range and hence $r_0 \in \mathrm{emb}(f(K), g(W))$. Thus, in the above infimum the set $\mathrm{emb}(f(K), g(W))$ is non-empty.

Before giving properties of $B_{K,W}(f, g)$, we briefly describe its interpretation and meaning. We denote by $\mathcal{P}(X)$ the set of probability measures over $X$. If the embedding gap between two functions is small, then $g$ is nearly an embedding of the range of $f$ into $\mathbb{R}^o$. $B_{K,W}(f, g)$ is constructed expressly to serve as an upper bound

$$\inf_{\mu_o \in \mathcal{P}(W)} \mathrm{W}_2\left(f_{\#}\mu_n, g_{\#}\mu_o\right) \leq B_{K,W}(f, g)$$

where $\mu_n \in \mathcal{P}(K)$ is given, and $\mathrm{W}_2(\nu_1, \nu_2)$ denotes the Wasserstein-2 distance with $\ell^2$ ground metric (Villani, 2008), as shown in Lemma 7 part 5. The above result has a simple meaning in the context of machine learning. Suppose we want to learn a generative model $g$ to (approximately) sample from a probability measure $\nu$ with low-dimensional support, by applying $g$ to samples from a *base* distribution $\mu_o$. Suppose further that $\nu$ is a pushforward of some (known or unknown) distribution $\mu_n$ via $f$. The embedding gap $B_{K,W}(f, g)$ then upper bounds the 2-Wasserstein distance between $\nu$ and $g_{\#}\mu_0$ for the best possible choice of $\mu_o$.[3]

---

[2]Note that if $X$ is a compact set, then continuity of the of $f^{-1} \colon f(X) \to X$ is automatic, and need not be assumed (Sutherland, 2009, Cor. 13.27). Moreover, if $f : \mathbb{R}^n \to \mathbb{R}^m$ is a continuous injective map that satisfies $|f(x)| \to \infty$ as $|x| \to \infty$, then by (Mukherjee, 2015, Cor. 2.1.23) the map $f^{-1} \colon f(\mathbb{R}^n) \to \mathbb{R}^n$ is continuous.

[3]The choice of $p$-Wasserstein distance is suitable for measures with mismatched low-dimensional support; this has been widely exploited in training generative models (Arjovsky et al., 2017).

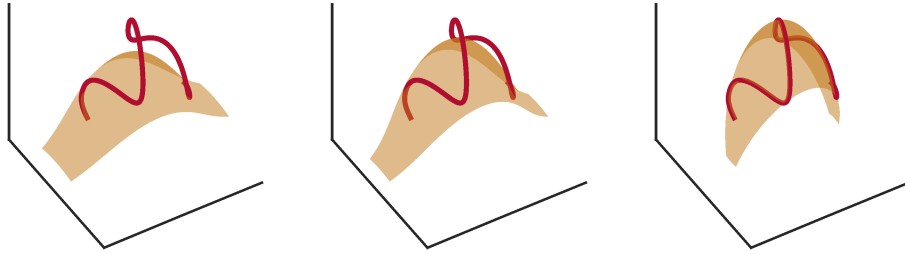

Figure 1: A visualization of the embedding gap. In all three figures we plot $f$ and $g_i$ for Left: $i = 1$, Center: $i = 2$ and Right: $i = 3$. Visually, we see that $g_i$ approaches $f$ as increases, and we compute $B_{K,W}(f, g_1) > B_{K,W}(f, g_2) > B_{K,W}(f, g_3) = 0$.

The embedding $r$ can be interpreted as a candidate transport map from any measure pushed forward by $f$, that can be pulled back through $g$. Loosely speaking, for $\mu_o' = g^{-1} \circ r \circ f_{\#}\mu_n$, $r$ is a valid Wasserstein transport map that transports $f_{\#}\mu_n$ to $g_{\#}\mu_o'$ with cost no more than $\|I - r\|_{L^\infty(f(K))}$. See Fig. 1 for a visualization of the embedding gap between two toy functions. Further, the embedding gap satisfies inequalities useful for studying networks of the form of Eqn. 1, see Lemma 7.

## 3.2 MANIFOLD EMBEDDING PROPERTY

We now introduce a central concept, the manifold embedding property (MEP). A family of networks satisfies the MEP if it nearly embeds a large class of manifolds of certain dimension and regularity, as measured by the embedding gap. The MEP is a property of a family of functions $\mathcal{E}^{o,m} \subset \mathrm{emb}(W, \mathbb{R}^m)$ where $W \subset \mathbb{R}^o$. For this manuscript, $\mathcal{E}^{o,m}$ will always be formed by taking $\mathcal{E}^{o,m} := \mathcal{T}^m \circ \mathcal{R}^{o,m}$.

We note here that the MEP is closely related to the question of whether or not a given $n$-dimensional manifold $\mathcal{M}$ that is an image of an embedding $f : K \to \mathbb{R}^m$, that is, $\mathcal{M} = f(K)$, $K \subset \mathbb{R}^n$, can be approximated by the images of neural networks $E : K \to \mathbb{R}^m$ of a given type. In particular, we will consider neural networks that are compositions where $R : \mathbb{R}^o \to \mathbb{R}^m$ are 'simple' non-universal functions, and flow-maps $T : \mathbb{R}^m \to \mathbb{R}^m$ that are diffeomorphism. This choice of applying non-universal expansive layers, and then diffromorphisms has some topological consequences, which we discuss below.

### TOPOLOGICAL OBSTRUCTIONS TO MANIFOLD LEARNING WITH NEURAL NETWORKS

We find that using non-universal expansive layers, followed by layers imposes some topological conditions on what can be approximated. We illustrate this with the following problem. When $n = 2$, $m = 3$, and $K = S^1 \subset \mathbb{R}^2$ is the circle. We consider maps $E = T \circ R : \mathbb{R}^n \to \mathbb{R}^m$ where $R : \mathbb{R}^n \to \mathbb{R}^m$ is, e.g., a linear map of rank $n$, and $T : \mathbb{R}^m \to \mathbb{R}^m$ is a coupling flow which is a homeomorphism. Let $f \in \mathrm{emb}(K, \mathbb{R}^3)$ be an embedding that maps $K$ to a trefoil knot $\mathcal{M} = f(S^1)$, see Fig. 2. Such a function $f$ can not be written as a restriction $T \circ R$ on $S^1$. In Sec. C.2.2 we prove this fact and build a related example where a measure, $\mu \in \mathcal{P}(\mathbb{R}^2)$, supported on an annulus is pushed forward to a measure supported on a knotted ribbon in $\mathbb{R}^3$ by an embedding $g : \mathbb{R}^2 \to \mathbb{R}^3$. For this measure, there are no $E := T \circ R$, with linear injective $R$ and embedding $T$, such that $g_{\#}\mu = E_{\#}\mu$.

To sidestep this fundamental difficulty, we define the MEP property for a certain subclass of manifolds $\{f(K) : f \in \mathcal{F}\}$. Finally, when considering flow networks which are universal approximators of $C^2$ diffeomorphisms, we restrict the class of manifolds to be approximated even further. This is because manifolds that are homeomorphic are not necessarily diffeomorphic; for example exotic spheres. These are topological structures that are homeomorphic, but not diffeomorphic, to the sphere Milnor (1956). Moreover, it is known that general homeomorphisms $F : \mathbb{R}^m \to \mathbb{R}^m$ can

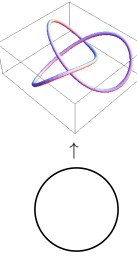

Figure 2: An illustration of the case when $n = 2$, $m = 3$, and $K = S^1$ is the circle. Here $f : S^1 \to \mathbb{R}^3$ is an embedding such that the curve $\mathcal{M} = f(S^1)$ is a trefoil knot. Due to knot theoretical reasons, there are no map $E = T \circ R : \mathbb{R}^2 \to \mathbb{R}^3$ such that $E(S^1) = \mathcal{M}$, where $R : \mathbb{R}^2 \to \mathbb{R}^3$ is a full rank linear map and $T : \mathbb{R}^3 \to \mathbb{R}^3$ is a homeomorphism. This shows that a combination of linear maps and coupling flow maps can not represent all embedded manifolds. For this reason, we define the class $\mathcal{I}(\mathbb{R}^n, \mathbb{R}^m)$ of extendable embeddings $f$. A similar 2-dimensional example can be obtained to a knotted ribbon, see Sec. C.2.2.

not be approximated in $C^0$-topology by $C^2$-smooth diffeomorphisms $T : \mathbb{R}^m \to \mathbb{R}^m$. See Müller (2014) for a precise statement. However, all $C^1$-smooth diffeomorphims $F : \mathbb{R}^m \to \mathbb{R}^m$ can be approximated in the strong topology of $C^1$ by $C^2$-smooth diffeomorphims $\tilde{F} : \mathbb{R}^m \to \mathbb{R}^m$, $\ell \geq k$, see Hirsch (2012), Ch. 2, Theorem 2.7. Because of this, we have to pay attention to the smoothness of the maps in the subset $\mathcal{F} \subset \mathrm{emb}(K, \mathbb{R}^m)$.

For these reasons, when we refer to the MEP, we consider it with respect to a class of functions $\mathcal{F} \subset \mathrm{emb}(\mathbb{R}^n, \mathbb{R}^m)$. The MEP can be interpreted as a density statement, saying that our networks $\mathcal{E}^{o,m}$ are dense in the '$B_{K,W}$ distance' in the set $\mathcal{F} \subset \mathrm{emb}(\mathbb{R}^n, \mathbb{R}^m)$. Two examples we are particularly interested in are the following. First, when $F = \Phi \circ A$ where $A : \mathbb{R}^n \to \mathbb{R}^m$ are linear maps of rank $n$ and $\Phi : \mathbb{R}^m \to \mathbb{R}^m$ are $C^k$ diffeomorphisms with $k \geq 1$. Second, when $F \in \mathrm{emb}(\mathbb{R}^n, \mathbb{R}^m)$.

**Definition 4.** Let $\mathcal{E}^{o,m} \subset \mathrm{emb}(\mathbb{R}^o, \mathbb{R}^m)$ and $\mathcal{F}^{n,m} \subset \mathrm{emb}(\mathbb{R}^n, \mathbb{R}^m)$ be two families of functions. We say that $\mathcal{E}^{o,m}$ has the $m, n, o$ Manifold Embedding Property (MEP) w.r.t. $\mathcal{F}^{n,m}$ if for every compact non-empty set $K \subset \mathbb{R}^n$, $f \in \mathcal{F}^{n,m}$, and $\epsilon > 0$, there is an $E \in \mathcal{E}^{o,m}$ and a compact set $W \subset \mathbb{R}^o$ such that the restriction of $f$ in $K$ and the restriction of $E$ in $W$ satisfy

$$B_{K,W}(f, E) < \epsilon. \tag{7}$$

When it is clear from the context, we abbreviate the $m, n, o$ MEP w.r.t. $\mathcal{F}^{n,m}$ simply by the $m, n, o$ MEP, or simply the MEP.

We also note here that if a model from $\mathbb{R}^o \to \mathbb{R}^m$ is a uniform universal approximator of $\mathcal{F}^{o,m} = C^0(\mathbb{R}^n, \mathbb{R}^m)$ on compact sets, such as those considered in Yarotsky (2017; 2018), then it has the $m, n, o$ MEP w.r.t $\mathcal{F}^{n,m}$ for any $n \leq o$. Thus, networks that are uniform universal approximators automatically possess the MEP, as shown in Lemma 1. However, we have a result that applies to network which are uniform universal approximators.

**Definition 5** (Uniform Universal Approximator). For a non-empty subset $\mathcal{F}^{n,m} \subset C(\mathbb{R}^n, \mathbb{R}^m)$, a family $\mathcal{E}^{n,m} \subset C(\mathbb{R}^n, \mathbb{R}^m)$ is said to be a uniform universal approximator of $\mathcal{F}^{n,m}$ if for every $f \in \mathcal{F}^{n,m}$, every non-empty compact $K \subset \mathbb{R}^n$, and each $\epsilon > 0$, there is an $E \in \mathcal{E}^{n,m}$ satisfying:

$$\sup_{x \in K} \|f(x) - E(x)\|_2 < \epsilon. \tag{8}$$

We also have the following function class, which is key to our analysis.

**Definition 6** (Extendable Embeddings). Let $\mathcal{I}(\mathbb{R}^n, \mathbb{R}^m)$ be the set of functions $F : \mathbb{R}^n \to \mathbb{R}^m$ that are compositions $F = \Phi \circ R$ where $R : \mathbb{R}^n \to \mathbb{R}^m$ is a linear map of rank $n$ and $\Phi : \mathbb{R}^m \to \mathbb{R}^m$ is a $C^1$-smooth diffeomorphism. We call $\mathcal{I}(\mathbb{R}^n, \mathbb{R}^m)$ be the set of extendable embeddings. We also denote $\mathcal{I}^k(\mathbb{R}^n, \mathbb{R}^m) = \mathcal{I}(\mathbb{R}^n, \mathbb{R}^m) \cap C^k(\mathbb{R}^n, \mathbb{R}^m)$. We observe that $\mathcal{I}(\mathbb{R}^n, \mathbb{R}^m) \subset \mathrm{emb}(\mathbb{R}^n, \mathbb{R}^m)$.

**Lemma 1.**    *(i) If $\mathcal{R}^{o,m}$ is a universal approximator of $C(\mathbb{R}^n, \mathbb{R}^m)$ and $I \in \mathcal{T}^m$ where $I$ is the identity map, then $\mathcal{E}^{o,m} := \mathcal{T}^m \circ \mathcal{R}^{o,m}$ has the MEP w.r.t. $\mathrm{emb}(\mathbb{R}^n, \mathbb{R}^m)$.*

   *(ii) If $\mathcal{R}^{o,m}$ is such that there is an injective $R \in \mathcal{R}^{o,m}$ and open set $U \subset \mathbb{R}^o$ such that $R|_{\overline{U}}$ is linear, and $\mathcal{T}^m$ is a* sup *universal approximator in the space of Diff$^2(\mathbb{R}^m, \mathbb{R}^m)$, in the sense of Teshima et al. (2020), of the $C^2$-smooth diffeomorphisms, then $\mathcal{E}^{o,m} := \mathcal{T}^m \circ \mathcal{R}^{o,m}$ has the MEP w.r.t. $\mathcal{F} = \mathcal{I}(\mathbb{R}^n, \mathbb{R}^m)$.*

The proof of Lemma 1 is in Appendix C.2.1. It has the following implications for the architectures studied in Section 2.

**Example 1.** Let $\mathcal{E} := \mathcal{T}^m \circ \mathcal{R}^{o,m}$, then

(i) If $\mathcal{T}^m$ is either (T1) or (T2) and $\mathcal{R}^{o,m}$ is (R4), then $\mathcal{E}^{o,m}$ has the $m, n, o$ MEP w.r.t. $\mathrm{emb}(\mathbb{R}^n, \mathbb{R}^m)$.

(ii) If $\mathcal{T}^m$ is (T2) with sigmoidal activations Huang et al. (2018), then if $\mathcal{R}^{o,m}$ is any of (R1), ..., (R4), then $\mathcal{E}^{o,m}$ has the $m, n, o$ MEP w.r.t. $\mathcal{I}(\mathbb{R}^n, \mathbb{R}^m)$.

The proof of Example 1 is in Appendix C.2.3.

We add a remark here showing that if $m$ is sufficiently larger than $n$, and subject to some regularity assumptions, The two classes considered in Example 1 coincide.

**Lemma 2.** *When $m \geq 3n + 1$ and $k \geq 1$, for any $C^k$ embedding $f \in \mathrm{emb}^k(\mathbb{R}^n, \mathbb{R}^m)$ and compact set $K \subset \mathbb{R}^n$, there is a map in the closures of the flow type neural network $E \in \mathcal{I}^k(\mathbb{R}^n, \mathbb{R}^m)$ such that $E(K) = f(K)$. Moreover,*

$$\mathcal{I}^k(K, \mathbb{R}^m) = \mathrm{emb}^k(K, \mathbb{R}^m) \tag{9}$$

This means that the topological structure of a manifold to approximate $\mathcal{M}$ in $\mathbb{R}^m$ is diffeomorphic to the set $K \subset \mathbb{R}^n$, and thus some flow type network can approximate $\mathcal{M}$.

The proof of Lemma 2 in Appendix C.2.4.

We finally have two more lemmas to present before moving on to our discussion of universality as it ties to the MEP.

**Lemma 3.** *Let $\mathcal{F}^{n,o} \subset \mathrm{emb}(\mathbb{R}^n, \mathbb{R}^o)$ and $\mathcal{F}^{o,m} \subset \mathrm{emb}(\mathbb{R}^0, \mathbb{R}^m)$ $\mathcal{E}_1^{p,o} \subset \mathrm{emb}(\mathbb{R}^p, \mathbb{R}^o)$ have the $o, n, p$ MEP w.r.t. $\mathcal{F}^{n,o} \subset \mathrm{emb}(\mathbb{R}^n, \mathbb{R}^o)$ and $\mathcal{E}_2^{o,m} \subset \mathrm{emb}(\mathbb{R}^o, \mathbb{R}^m)$ have the $m, o, o$ MEP w.r.t. $\mathcal{F}^{o,m} \subset \mathrm{emb}(\mathbb{R}^o, \mathbb{R}^m)$. If each $E_2^{o,m} \in \mathcal{E}_2^{o,m}$ is locally Lipschitz, then $\mathcal{E}_2^{o,m} \circ \mathcal{E}_1^{p,o}$ has the $m, n, p$ MEP w.r.t. $\mathcal{F}^{o,m} \circ \mathcal{F}^{n,o}$.*

The proof of Lemma 3 is in Appendix C.2.5.

We note that when the elements of $\mathcal{E}_2^{o,m}$ are differentiable, local Lipschitzness is automatic, and need not be assumed, see e.g. (Tao, 2009, Ex. 10.2.6). We also record a near-converse (proved in C.2.6) of Lemma 3 that shows that if $\mathcal{E}_2^{o,m} \circ \mathcal{E}_1^{p,o}$ has the $m, n, p$ MEP, then $\mathcal{E}_2^{o,m}$ has the $m, n, o$ MEP.

**Lemma 4.** *Let $\mathcal{F} \subset \mathrm{emb}(\mathbb{R}^n, \mathbb{R}^m)$. Let $\mathcal{E}_1^{p,o} \subset \mathrm{emb}(\mathbb{R}^p, \mathbb{R}^o)$ and $\mathcal{E}_2^{o,m} \subset \mathrm{emb}(\mathbb{R}^o, \mathbb{R}^m)$ be such that $\mathcal{E}_2^{o,m} \circ \mathcal{E}_1^{p,o}$ has the $m, n, p$ MEP with respect to family $\mathcal{F}$. Then $\mathcal{E}_2^{o,m}$ has the $m, n, o$ MEP with respect to family $\mathcal{F}$.*

### 3.3 UNIVERSALITY

We now present our universal approximation result for networks given in Eqn. Eq. 1 and a decoupling property.

**Theorem 1** (Qualitative Universality for Embeddings). *Let $n_0 = n, n_L = m$, $\mu \in \mathcal{P}(K)$ be an absolutely continuous measure w.r.t. Lebesgue measure. Further let, for each $\ell = 1, \ldots, L$, $\mathcal{E}_\ell^{n_{\ell-1}, n_\ell} := \mathcal{T}_\ell^{n_\ell} \circ \mathcal{R}_\ell^{n_{\ell-1}, n_\ell}$ where $\mathcal{R}_\ell^{n_{\ell-1}, n_\ell}$ is a family of injective expansive elements that contains a linear map, and $\mathcal{T}_\ell^{n_\ell}$ is a family of bijective family networks. Finally let $\mathcal{T}_0^n$ be distributionally universal, i.e. for any absolutely continuous $\mu \in \mathcal{P}(\mathbb{R}^n)$ and $\nu \in \mathcal{P}(\mathbb{R}^n)$, there is a $\{T_i\}_{i=1}^\infty$ such that $T_{i\#}\mu \to \nu$ in distribution. Suppose that one of the following two cases hold:*

*(i) Let $F \in \mathcal{F}_L^{n_{L-1}, m} \circ \cdots \circ \mathcal{F}_1^{n, n_1}$ and $\mathcal{E}_\ell^{n_{\ell-1}, n_\ell}$ have the the $n_\ell, n, n_{\ell-1}$ MEP for $\ell = 1, \ldots, L$ with respect to $\mathcal{F}_\ell^{n_{\ell-1}, n_\ell}$.*

*(ii) Let $F \in \mathrm{emb}^1(\mathbb{R}^n, \mathbb{R}^m)$ be a $C^1$-smooth embedding, and assume that for $\ell = 1, \ldots, L$ it holds that $n_\ell \geq 3n_{\ell-1} + 1$ and the families $\mathcal{T}_\ell^{n_\ell}$ are dense in the space of $C^2$-diffeomorphism $\mathrm{Diff}^2(\mathbb{R}^{n_\ell})$.*

*Then, there is a sequence of $\{E_i\}_{i=1,\ldots,\infty} \subset \mathcal{E}_L^{n_{L-1}, m} \circ \cdots \circ \mathcal{E}_1^{n_1, n} \circ \mathcal{T}_0^n$ such that*

$$\lim_{i \to \infty} \mathrm{W}_2 \left( F_{\#}\mu, E_{i\#}\mu \right) = 0. \tag{10}$$

The proof of Theorem 1 is in Appendix C.3.1. As discussed in above and in Figure 2, there are topological obstructions for Theorem 1 with a general embedding $F : \mathbb{R}^n \to \mathbb{R}^m$. When $n = 2$,

$m = 3$, $L = 1$, and $\mu$ is the uniform measure on an annulus $K \subset \mathbb{R}^2$ target measure $F_{\#}\mu$ is the uniform measure on a knotted ribbon $\mathcal{M} = F(K) \subset \mathbb{R}^3$. There are no injective linear maps $R : \mathbb{R}^2 \to \mathbb{R}^3$ and diffeomorphisms $T : \mathbb{R}^3 \to \mathbb{R}^3$ such that $E = T \circ R$ would satisfy $\mathcal{M} = E(K)$ and $E_{\#}\mu = F_{\#}\mu$. This happen even though the set of diffeomorphism $T : \mathbb{R}^3 \to \mathbb{R}^3$ are universal distributional approximators. In this case the condition $m = n_1 \geq 3n + 1$ is not valid.

We remark here that our networks are designed expressly to approximate manifolds, and hence injectivity is key. This separates our results from, e.g. (Lee et al., 2017, Theorem 3.1) or (Lu & Lu, 2020, Theorem 2.1), where universality results of ReLU networks are also obtained.

The previous theorem shows that the entire network is universal if it can be broken into pieces that have the MEP. The following lemma, proved in Appendix C.3.2, shows that if $\mathcal{E}^{n,m} = \mathcal{H}^{o,m} \circ \mathcal{G}^{n,o}$, then $\mathcal{H}^{o,m}$ must have the $m, n, o$ MEP if $\mathcal{E}^{n,m}$ is universal.

**Lemma 5.** *Suppose that* $\mathcal{E}^{n,m} = \mathcal{H}^{o,m} \circ \mathcal{G}^{n,o}$ *where* $\mathcal{E}^{n,m} \subset \mathrm{emb}(\mathbb{R}^n, \mathbb{R}^m)$, $\mathcal{H}^{o,m} \subset \mathrm{emb}(\mathbb{R}^o, \mathbb{R}^m)$, *and* $\mathcal{G}^{n,o} \subset \mathrm{emb}(\mathbb{R}^n, \mathbb{R}^o)$. *If* $\mathcal{H}^{o,m}$ *does not have the* $m, n, o$ *MEP w.r.t.* $\mathcal{F}$, *then there exists a* $f \in \mathcal{F}$, *compact* $K \subset \mathbb{R}^n$ *and* $\epsilon > 0$ *such that for all* $E \in \mathcal{E}^{n,m}$, *and* $r \in \mathrm{emb}(f(K), E(W))$

$$\|I - r\|_{L^{\infty}(K)} \geq \epsilon. \tag{11}$$

The proof of Theorem 1 also implies that, loosely speaking, later layers decouple from earlier ones. That is, given a sequence of functions that has the MEP on the last $L - \tilde{\ell}$ layers, there are always functions in the first $\tilde{\ell}$ layers so that the entire network is end-to-end optimal.

**Corollary 1.** *Let* $\mathcal{F}^{n,o} \subset \mathrm{emb}(\mathbb{R}^n, \mathbb{R}^o), \mathcal{F}^{o,m} \subset \mathrm{emb}(\mathbb{R}^o, \mathbb{R}^m)$, *and let* $\mathcal{E}^{o,m} \subset \mathrm{emb}(\mathbb{R}^o, \mathbb{R}^m)$ *have the* $m, n, o$ *MEP w.r.t.* $\mathcal{F}^{o,m} \circ \mathcal{F}^{n,o}$. *For every* $F \in \mathcal{F}^{o,m} \circ \mathcal{F}^{n,o}$ *then there is a compact* $K \subset \mathbb{R}^n$ *and* $\{E_i\}_{i=1}^{\infty} \subset \mathcal{E}^{o,m}$ *such that*

$$\lim_{i \to \infty} B_{K,W}(F, E_i) = 0. \tag{12}$$

*Further, if* $\mathcal{E}^{n,o} \subset \mathrm{emb}(W', \mathbb{R}^o)$ *has the* $o, n, n$ *MEP w.r.t.* $\mathcal{F}^{n,o}$, *and* $\mathcal{T}^n$ *is a universal approximator for distributions, then for any* $\mu \in \mathcal{P}(K)$ *where* $K \subset \mathbb{R}^n$ *is compact, there is a sequence* $\{E_i'\}_{i=1,\dots,\infty} \subset \mathcal{E}^{n,o}$ *and* $\{T_i\}_{i=1,\dots,\infty} \subset \mathcal{T}^n$ *so that*

$$\lim_{i \to \infty} W_2 \left( F_{\#}\mu, E_i \circ E_i' \circ T_{i\#}\mu \right) = 0. \tag{13}$$

The proof of Corollary 1 is in Appendix C.3.3. Approximation results for neural networks are typically given in terms of the network end-to-end. Corollary 1 shows that the layers of approximating networks can in fact be built one at a time. It is related to an observation made in (Brehmer & Cranmer, 2020, Section B) about training strategies, where the authors remark that they 'expect faster and more robust training of a network' of the form in Eqn. 1 when $L = 1$, that is $\mathcal{F} = \mathcal{T}_1^m \circ \mathcal{R}_1^{n,m} \circ \mathcal{T}_0^n$. Corollary 1 shows that there exists a minimizing sequence in $\mathcal{T}_1^m$ that need only minimize Eqn. 12; the $\mathcal{T}_0^n$ layers can be minimized after. We can further combine Lemma 5 and Cor. 1 to prove that not only can the network from Brehmer & Cranmer (2020) be trained layerwise, but that any universal network can necessarily be trained layerwise in reverse order, provided that it can be written as a composition of two smaller layers.

### 3.4 LAYER-WISE INVERSION, UNCERTAINTY QUANTIFICATION AND RECOVERY OF WEIGHTS

In this subsection, we describe how our network can be augmented with more useful properties if the architecture satisfies a few more assumptions without affecting universal approximation. We focus on a new layerwise projection result, with a further discussion of Bayesian uncertainty quantification, and black-box recovery of our network's weights in Appendices D.2 & D.3.

Given a point $y \in \mathbb{R}^m$ that does not lie in the range of the network, projecting $y$ onto the range of the network is a practical problem without an obvious answer. The crux of the problem is inverting the injective (but non-invertible) $\mathcal{R}$ layers when $\mathcal{R}$. When $\mathcal{R}$ contains only full-rank matrices as in (R1) or (R2) then we can compute a least-squares solution. If, however, $\mathcal{R}$ contains layers which are only piecewise linear, as in (R3), then the problem of computing a least squares solution is more difficult, see Fig. 3. Nevertheless, we find that if $\mathcal{R}$ is (R3) we can still compute a least-squares solution.

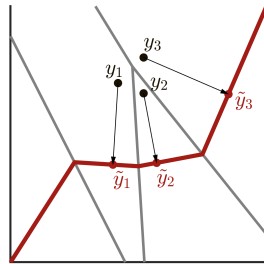

Figure 3: A schematic showing that, for a toy problem, the least-squares projection to a piecewise affine range can be discontinuous. Left: A partitioning of $\mathbb{R}^2$ into classes with gray boundaries. Two points $y, y'$ are in the same class if they are both closest to the same affine piece of $R(\mathbb{R})$, the range of $R$. The three points $y_1, y_2$ and $y_3$ are each projected to the closest three points on $R(\mathbb{R})$ yielding $\tilde{y}_1, \tilde{y}_2$ and $\tilde{y}_3$. Note that the projection operation is continuous within each section, but discontinuous across gray boundaries between section.

**Assumption 1.** *Let $\mathcal{R}$ be given by one of (R1) or (R2), or else (R3) when $m = 2n$.*

If $\mathcal{R}$ only contains linear operators, then the least-squares problem can be computed by solving the normal equations (see (Golub, 1996, Section 5.3).) This includes cases (R1) or (R2). For (R3) we have the following result for $D = I^{n \times n}$ and $M$ is an empty matrix.

**Definition 7.** Let $W = \begin{bmatrix} B^t & -DB^t \end{bmatrix}^t \in \mathbb{R}^{2n \times n}$ and $y \in \mathbb{R}^{2n}$ be given, and let $R(x) = \mathrm{ReLU}(Wx)$. Then define

$$c(y) \in \mathbb{R}^{2n} \quad c(y) := \max\left( \begin{bmatrix} I^{n \times n} & -I^{n \times n} \\ -I^{n \times n} & I^{n \times n} \end{bmatrix} y, 0 \right) \tag{14}$$

$$\Delta_y \in \mathbb{R}^{n \times n} \quad [\Delta_y]_{i,i} := \begin{cases} 0 & \text{if } [c(y)]_{i+n} = 0 \\ 1 & \text{if } [c(y)]_{i+n} > 0 \end{cases} \quad [\Delta_y]_{ij} = 0 \text{ if } i \neq j \tag{15}$$

$$M_y \in \mathbb{R}^{n \times 2n} \quad M_y := \begin{bmatrix} (I^{n \times n} - \Delta_y) & \Delta_y \end{bmatrix} \tag{16}$$

where the max in Eqn. 14 is taken element-wise.

**Theorem 2.** *Let $y \in \mathbb{R}^{2n}$. If for $i = 1, \ldots, n$, $[y]_i \neq [y]_{i+n}$ then*

$$R^\dagger(y) := (M_y W)^{-1} M_y y = \operatorname*{argmin}_{x \in \mathbb{R}^n} \|y - R(x)\|_2. \tag{17}$$

*Further, if there is a $i \in \{1, \ldots, n\}$ such that $[y]_i = [y]_{i+n}$, then there are multiple minimizers of $\|y - R(x)\|_2$, one of which is $R^\dagger(y)$.*

The proof of Theorem 2 is given in Appendix D.1.

*Remark 2.* We note that Theorem 2 is different from many of the existing work on inverting expansive layers, e.g. Aberdam et al. (2020); Bora et al. (2017); Lei et al. (2019), our result gives a direct inversion algorithm that is provably the least-squares minimizer. Further, if each expansive layer is any combination of (R1), (R2), or (R3) then the entire network can be inverted end-to-end by using either the above result or solving the normal equations directly.

## 4 CONCLUSION

Bijective flow networks are a powerful tool for learning a push-forward mappings in a space of fixed dimension. Increasingly, these flow networks have been used in combination with networks that increase dimension in order to produce networks which are purportedly universal.

In this work, we have studied the theory underpinning these flow and expansive networks by introducing two new notions, the embedding gap and the manifold embedding property. We show that these notions are both necessary and sufficient for proving universality, but require important topological and geometrical considerations which are, heretofore, under-explored in the literature. We also find that optimality of the studied networks can be established 'in reverse,' by minimizing the embedding gap, which we expect opens the door to convergence of layer-wise training schemes. Without compromising universality, we can also use specific expansive layers with a new layer-wise projection result. Moreover, we show that the studied networks provide Bayesian uncertainty quantification and allow black-box recovery of their weights.

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

## APPENDIX A    SUMMARY OF NOTATION

Throughout the paper we make heavy use of the following notation.

1. Unless otherwise stated, $X$ and $Y$ always refer to subsets of Euclidean space, and $K$ and $W$ always refer to compact subsets of Euclidian space.
2. $f \in C(X, Y)$ means that $f \colon X \to Y$ is continuous.
3. For families of functions $\mathcal{F}$ and $\mathcal{G}$ where each $\mathcal{F} \ni f \colon X \to Y$ and $\mathcal{G} \ni g \colon Y \to Z$, then we define $\mathcal{G} \circ \mathcal{F} = \{g \circ f \colon X \to Z \colon f \in \mathcal{F},\ g \in \mathcal{G}\}$.
4. $f \in \mathrm{emb}(X, Y)$ means that $f \in C(X, Y)$ is continuous and injective on the range of $f$, i.e. an embedding, and furthermore that $f^{-1} \colon f(X) \to X$ is continuous.
5. $\mu \in \mathcal{P}(X)$ means that $\mu$ is a probability measure over $X$.
6. $\mathrm{W}_2(\mu, \nu)$ for $\mu, \nu \in \mathcal{P}(X)$ refers to the Wasserstein-2 distance, always with $\ell_2$ ground metric.
7. $\|\cdot\|_{L^p(X)}$ refers to the $L^p$ norm of functions, from $X$ to $\mathbb{R}$.
8. For vector-valued $f \colon X \to Y$, $\|f\|_{L^\infty(X)} = \mathrm{ess\,sup}_{x \in \mathcal{X}} \|f\|_2$. Note that $Y$ is always finite dimensional, and so all discrete $1 \le q \le \infty$ norms are equivalent.
9. $\mathrm{Lip}(g)$ refers to the Lipschitz constant of $f$.
10. For $x \in \mathbb{R}^n$, $[x]_i \in \mathbb{R}$ is the $i$'th component of $x$. Similarly, for matrix $A \in \mathbb{R}^{m \times n}$, $[A]_{ij}$ refers to the $j$'th element in the $i$'th column.

## APPENDIX B    DETAILED COMPARISON TO PRIOR WORK

### B.1    CONNECTION TO BREHMER & CRANMER (2020)

In Brehmer & Cranmer (2020), the authors introduce manifold-learning flows as an invertible method for learning probability density supported on a low-dimensional manifold. Their model can be written as

$$\mathcal{F} = \mathcal{T}_1^m \circ \mathcal{R}^{n,m} \circ \mathcal{T}_0^n \tag{18}$$

where $\mathcal{T}_1^m \subset C(\mathbb{R}^m, \mathbb{R}^m)$, $\mathcal{T}_0^m \subset C(\mathbb{R}^n, \mathbb{R}^n)$, and $\mathcal{R} = \left\{ \begin{bmatrix} I^{n \times n} \\ \mathbf{0}^{(m-n) \times n} \end{bmatrix} \right\}$ is a zero-padding (R1).

They invert $F \in \mathcal{F}$ in two different ways. For manifold-learning flows ($\mathcal{M}$-flows) they restrict $\mathcal{T}_1^m$ to be an invertible flow, and for manifold-learning flows with separate encoder ($\mathcal{M}_e$-flows) they place no such restrictions on $\mathcal{T}_1^m$ and instead train a separate neural network $e$ to invert elements of $\mathcal{T}_1^m$.

Our results apply out-of-the-box to the architectures used in Experiment A of Brehmer & Cranmer (2020). The architecture described in Eqn. 18 is of the form of Eqn. 1 where $L = 1$. Further, although they are not studied here, our analysis can also be applied to quadratic flows.

The network used in (Brehmer & Cranmer, 2020, Experiment 4.A) uses coupling networks, (T1), where $\mathcal{T}_1^m$ and $\mathcal{T}_0^n$ are both 5 layers deep. For (Brehmer & Cranmer, 2020, Experiments 4.B and 4.C) the authors choose expressive elements $\mathcal{T}$ that are rational quadratic flows Durkan et al. (2019b) for both $\mathcal{T}_1^m$ and $\mathcal{T}_0^n$. In Experiment 4.B they let $T_1$ and $T_0$ again be 5 layers deep, and in 4.C they again let $T_1$ by 20 layers deep and $T_0$ 15 layers. For the final experiment, 4.D, the choose more complicated expressive elements that combine Glow Kingma & Dhariwal (2018) and Real NVP Dinh et al. (2016) architectures. These elements include the actnorm, $1 \times 1$ convolutions and rational-quadratic coupling transformations along with a multi-scale transformation.

The authors mention universality of their network without our proof, but our universality results in Theorem 1 apply to their networks from Experiment A wholesale. Further in their work the authors describe how training can be split into a manifold phase and density phase, wherein the manifold phase $\mathcal{T}_1^m$ is trained to learn the manifold, and in the density phase $\mathcal{T}_1^m$ if fixed and $\mathcal{T}_0^n$ is trained to learn the density thereupon. This statement is made formal and proven by our Cor. 1.

### B.2 Connection to Kothari et al. (2021)

In Kothari et al. (2021), the authors introduce the 'Trumpet' architecture, for it's architecture, which has many alternating flow networks & expansive layers with many flow-networks in the low-dimensional early stages of the network, which gives the architecture a shape similar to the titular instrument.

The architecture studied in Kothari et al. (2021) is precisely of the form of Eqn. 1, where the bijective flow networks are revnets Gomez et al. (2017); Jacobsen et al. (2018) architecture, and the expansive elements are $1 \times 1$ convolutions, as in (R2). To out knowledge, there are no results that show that the revnets used are universal approximators, but if they revnets are substituted with either (T1) or (T2), then the, we could apply Theorem 1 to the resulting architecture.

The authors of Kothari et al. (2021) also remark that their entire network can be designed in order to facilitate uncertainty quantification, a remark that we too can apply to our networks, as discussed further in Section D.2.

## Appendix C  Proofs

### C.1  Main Results

#### C.1.1  Helper Lemma

Before presenting the proof of Lemma 7, we present the following three helper inequalities

**Lemma 6.** *For all of the following results, $f \in \mathrm{emb}(K, \mathbb{R}^m)$ and $g \in \mathrm{emb}(W, \mathbb{R}^m)$ and $n \le o \le m$.*

*1.*

$$B_{K,W}(f,g) \ge \sup_{x_n \in K} \inf_{x_o \in W} \| g(x_o) - f(x_n) \|_2 . \tag{19}$$

*2. Let $X, Y \subset W$, let $g$ be Lipschitz on $W$, and $r \in \mathrm{emb}(X, Y)$. Then, there is a $r' \in \mathrm{emb}(g(X), g(Y))$ such that $g \circ r = r' \circ g$ and $\| I - r' \|_{L^\infty(g(X))} \le \| I - r \|_{L^\infty(X)} \mathrm{Lip}(g)$.*

*3.*

$$\|I - r\|_{L^\infty(K)} = \|I - r^{-1}\|_{L^\infty(r(K))} \tag{20}$$

*4. Let $K \subset \mathbb{R}^n$, $X \subset \mathbb{R}^p$ and $W \subset \mathbb{R}^o$ be compact sets. Also, let $f \in \mathrm{emb}(K, W)$ and $h \in \mathrm{emb}(X, W)$, and let $g \in \mathrm{emb}(W, \mathbb{R}^m)$ be a Lipschitz map. Then*

$$B_{K,X}(g \circ f, g \circ h) \leq \mathrm{Lip}(g) B_{K,X}(f, h). \tag{21}$$

*Proof.*     1. Let $r \in C(f(K), g(W))$, then

$$
\begin{aligned}
\|I - r\|_{L^\infty(f(K))} &= \sup_{x_n \in K} \|(I - r)f(x_n)\|_2 = \sup_{x_n \in K} \|f(x_n) - r \circ f(x_n)\|_2 \\
&= \sup_{x_n \in K} \|f(x_n) - g(x_o)\|_2 \ \text{where } x_o = g^{-1} \circ r \circ f(x_n) \\
&\geq \sup_{x_n \in K} \inf_{x_o \in W} \|f(x_n) - g(x_o)\|_2 \, .
\end{aligned}
$$

2. $g$ is injective on $X$, hence we can define $r'$ such that $r' = g \circ r \circ g^{-1} : g(X) \to g(r(X)) \subset g(Y)$ such that $r' \in \mathrm{emb}(g(X), g(Y))$, and thus $\forall x \in X$,

$$\|(I - r') \circ g(x)\|_2 = \|g(x) - g \circ r(x)\|_2 \leq \mathrm{Lip}(g) \|I - r\|_{L^\infty(X)} \tag{22}$$

where we have used $\|r(x) - x\|_2 \leq \|I - r\|_{L^\infty(X)}$.

3. For every $x \in r(K)$, we have a $y \in K$ such that $x = r(y)$, thus $\forall x \in r(K)$,

$$\left\|\left(I - r^{-1}\right)(x)\right\|_2 = \|(r - I)(y)\|_2 . \tag{23}$$

But $r$ is clearly surjective onto it's range, hence taking the supremum over all $x \in X$ yields

$$\left\|I - r^{-1}\right\|_{L^\infty(r(K))} = \|I - r\|_{L^\infty(K)} \tag{24}$$

4. As $g \in \mathrm{emb}(W, \mathbb{R}^m)$, the map $g : W \to g(W)$ is a homeomorphism and there is $g^{-1} \in \mathrm{emb}(g(W), W)$. For a map $r \in \mathrm{emb}(g \circ f(K), g \circ h(X))$, we see that $\hat{r} = g^{-1} \circ r \circ g \in \mathrm{emb}(f(K), h(X))$. Also, the opposite is valid as if $\hat{r} \in \mathrm{emb}(f(K), h(X))$ then $r = g \circ \hat{r} \circ g^{-1} \in \mathrm{emb}(g \circ f(K), g \circ h(X))$. Thus

$$
\begin{aligned}
B_{K,X}(g \circ f, g \circ h) &= \inf_{r \in \mathrm{emb}(g \circ f(K), g \circ h(X))} \|I - r\|_{L^\infty(g \circ f(K))} \\
&= \inf_{r = g \circ \hat{r} \circ g^{-1} \in \mathrm{emb}(g \circ f(K), g \circ h(X))} \|I - g \circ \hat{r} \circ g^{-1}\|_{L^\infty(g \circ f(K))} \\
&= \inf_{\hat{r} \in \mathrm{emb}(f(K), h(X))} \|g \circ (I - \hat{r}) \circ g^{-1}\|_{L^\infty(g \circ f(K))} \\
&\leq \mathrm{Lip}(g) \inf_{\hat{r} \in \mathrm{emb}(f(K), h(X))} \|(I - \hat{r}) \circ g^{-1}\|_{L^\infty(g \circ f(K))} \\
&\leq \mathrm{Lip}(g) \inf_{\hat{r} \in \mathrm{emb}(f(K), h(X))} \|I - \hat{r}\|_{L^\infty(f(K))} \\
&\leq \mathrm{Lip}(g) B_{K,X}(f, h)
\end{aligned}
$$

$\square$

### C.1.2    Lemma 7, Useful Embedding Gap Inequalities

**Lemma 7.** *Let $f \in \mathrm{emb}(K, \mathbb{R}^m)$ and $g \in \mathrm{emb}(W, \mathbb{R}^m)$ and $n \leq o \leq m$. Then the following hold:*

*1. $B_{K,W}(f, g) \leq \sup_{x \in K} \|g \circ h(x) - f(x)\|_2$ where $h \in \mathrm{emb}(K, \mathbb{R}^o)$ is a map satisfying $h(K) \subset W$.*

*2. For any $X$ that is the closure of an open set , if $h \in \mathrm{emb}(X, W)$ then*

$$B_{K,W}(f, g) \leq B_{K,X}(f, g \circ h) \tag{25}$$

3. *For any $r \in \mathrm{emb}(f(K), \mathbb{R}^m)$,*

$$B_{K,W}(f,g) \leq \|I - r\|_{L^\infty(f(K))} + B_{K,W}(r \circ f, g). \tag{26}$$

4. *For any $r \in \mathrm{emb}(f(K), g(W))$ and $h \in \mathrm{emb}(X, W)$ where $X \subset \mathbb{R}^p$ is the closure of a compact set where $n \leq p \leq o$ we have that*

$$B_{K,X}(f, g \circ h) \leq \|I - r\|_{L^\infty(f(K))} + \mathrm{Lip}(g) B_{K,X}(g^{-1} \circ r \circ f, h) \tag{27}$$

*where $\mathrm{Lip}(g)$ denotes the Lipschitz constant of g.*

5. *For any $\mu_n \in \mathcal{P}(K)$, there is a $\mu_o \in \mathcal{P}(W)$ such that*

$$\mathrm{W}_2(f_\# \mu_n, g_\# \mu_o) \leq B_{K,W}(f,g). \tag{28}$$

*The proof of Lemma 7.*      1. If we let $r := g \circ h \circ f^{-1}$, then $r \in \mathrm{emb}(f(K), g(W))$, and

$$B_{K,W}(f,g) \leq \left\| \|(I - r) \circ f(x)\|_2 \right\|_{L^\infty(K)} \tag{29}$$

$$= \left\| \|f(x) - g \circ h \circ f^{-1} \circ f(x)\|_2 \right\|_{L^\infty(K)} \leq \sup_{x \in K} \|f(x) - g \circ h(x)\|_2. \tag{30}$$

2. Given that $g \circ h(X) \subset g(W)$, we have that $\mathrm{emb}(f(K), g \circ h(X)) \subset \mathrm{emb}(f(K), g(W))$, thus the infimum in Eqn. 6 is taken over a smaller set, thus $B_{K,W}(f,g) \leq B_{K,X}(f, g \circ h)$.

3. Note that for any $r' \in \mathrm{emb}(r \circ f(K), g(W))$, $r' \circ r \in \mathrm{emb}(f(K), g(W))$, and so we have

$$B_{K,W}(f,g) \leq \|I - r' \circ r\|_{L^\infty(f(K))} \leq \|I - r\|_{L^\infty(f(K))} + \|r - r' \circ r\|_{L^\infty(f(K))} \tag{31}$$

$$= \|I - r\|_{L^\infty(f(K))} + \|I - r'\|_{L^\infty(r \circ f(K))} \tag{32}$$

where we have used that $r$ is injective for the final equality. This holds for all possible $r'$, hence we have the result.

4. Recall that $f \in \mathrm{emb}(K, W)$, $g \in \mathrm{emb}(W, \mathbb{R}^m)$, $h \in \mathrm{emb}(X, W)$ and $r \in \mathrm{emb}(f(K), g(W))$. Then $g^{-1} \in \mathrm{emb}(g(W), W)$. As $r \circ f(K) \subset g(W)$, we see that

$$r \circ f = g \circ g^{-1} \circ r \circ f.$$

This, Lemma 7, point 3 and Lemma 6, point 4 yield that

$$B_{K,X}(f, g \circ h) \leq \|I - r\|_{L^\infty(f(K))} + B_{K,X}(r \circ f, g \circ h)$$
$$\leq \|I - r\|_{L^\infty(f(K))} + B_{K,X}(g \circ g^{-1} \circ r \circ f, g \circ h)$$
$$\leq \|I - r\|_{L^\infty(f(K))} + \mathrm{Lip}(g) B_{K,X}(g^{-1} \circ r \circ f, h),$$

which proves the claim.

5. Let $r \in \mathrm{emb}(f(K), g(W))$ be such that $\|I - r\|_{L^\infty(\mathrm{Range}(f))} \leq B_{K,W}(f,g) + \epsilon$ for some $\epsilon > 0$, then for every $x \in K$, there exists $y \in W$ such that $g(y) = r \circ f(x)$. From injectivity of $g$, we have that $y = g^{-1} \circ r \circ f(x)$. Note that $g^{-1} \circ r \circ f \in \mathrm{emb}(K, W)$, hence $K' := g^{-1} \circ r \circ f(K)$ is compact. Define $\mu' := (g^{-1} \circ r \circ f)_\# \mu$. Clearly $g_\# \mu' = r \circ f_\# \mu$, and thus

$$\mathrm{W}_2(g_\# \mu', f_\# \mu) \leq \mathrm{W}_2(g_\# \mu', r \circ f_\# \mu) + \mathrm{W}_2(r \circ f_\# \mu, f_\# \mu) \tag{33}$$

but $\mathrm{W}_2(g_\# \mu', r \circ f_\# \mu) = 0$ and

$$\mathrm{W}_2(r \circ f_\# \mu, f_\# \mu) \leq \left( \int_K \|I - r\|_2^2 \, df_\# \mu \right)^{1/2} \leq B_{K,W}(f,g) + \epsilon. \tag{34}$$

This is true for every $\epsilon > 0$, hence taking the infimum over all $r \in \mathrm{emb}(f(K), g(K))$ establishes $\mathrm{W}_2(g_\# \mu', f_\# \mu) \leq B_{K,W}(f,g) + \epsilon$ for every $\epsilon > 0$, and thus we have that $\mathrm{W}_2(g_\# \mu', f_\# \mu) \leq B_{K,W}(f,g)$.

$\square$

## C.2 Manifold Embedding Property

### C.2.1 The proof of Lemma 1

*The proof of Lemma 1.*      (i) Let us consider $\epsilon > 0$, a compact set $K \subset \mathbb{R}^n$ and $f \in \mathrm{emb}(\mathbb{R}^n, \mathbb{R}^m)$. Let $W = K \times \{0\}^{o-n}$ and $F : \mathbb{R}^o \to \mathbb{R}^m$ be the map given by $F(x, y) = f(x)$, $(x, y) \in \mathbb{R}^n \times \mathbb{R}^{o-n}$. Because $\mathcal{R}^{o,m}$ is a uniform universal approximator of $C(\mathbb{R}^n, \mathbb{R}^m)$, there is an $R \in \mathcal{R}^{o,m}$ such that $\|F - R\|_{L^\infty(W)} < \epsilon$. Then for the map $E = I \circ R$ we have that $B_{K,W}(f, E) < \epsilon$. This is true for every $\epsilon > 0$, and so $\mathcal{E}^{o,m}$ has the MEP property w.r.t. the family $\mathrm{emb}(\mathbb{R}^n, \mathbb{R}^m)$.

  (ii) Recall that $f := \Phi_0 \circ R_0$ for $\Phi_0 \in \mathrm{Diff}^1(\mathbb{R}^m, \mathbb{R}^m)$ and linear $R_0 : \mathbb{R}^n \to \mathbb{R}^m$, and that $R \in \mathcal{R}$ is such that $R|_{\overline{U}}$ is linear for open $U$. We present the proof in the case when $n = o$, and we make the assumption that $R|_K$ is linear. In this case, we have the existence of an affine map $A : \mathbb{R}^m \to \mathbb{R}^m$ so that $R_0 = A \circ R$ so that $\tilde{K} := R_0(K) = A(R(K))$. Let $\epsilon > 0$ be given. By (Hirsch, 2012, Chapter 2, Theorem 2.7), the space $\mathrm{Diff}^2(\mathbb{R}^m, \mathbb{R}^m)$ is dense in the space $\mathrm{Diff}^1(\mathbb{R}^m, \mathbb{R}^m)$, and so there is some $\Phi_1 \in \mathrm{Diff}^2(\mathbb{R}^m, \mathbb{R}^m)$ such that

$$\|\Phi_1|_{\tilde{K}} - \Phi_0|_{\tilde{K}}\|_{L^\infty(\tilde{K};\mathbb{R}^m)} < \frac{\epsilon}{2}.$$

Then, let $T \in \mathcal{T}^m$ be such that $\|T - \Phi_1 \circ A\|_{L^\infty(R(K);\mathbb{R}^m)} < \frac{\epsilon}{2}$. Then we have that

$$
\begin{aligned}
\|T \circ R - f\|_{L^\infty(K)} &= \|T \circ R - \Phi_0 \circ R_0\|_{L^\infty(K)} \\
&\le \|T \circ R - \Phi_1 \circ A \circ R\|_{L^\infty(K)} + \|\Phi_1 \circ A \circ R - \Phi_0 \circ R_0\|_{L^\infty(K)} \\
&\le \|T - \Phi_1 \circ A\|_{L^\infty(R(K))} + \|\Phi_1 \circ A \circ R - \Phi_0 \circ A \circ R\|_{L^\infty(K)} \\
&< \frac{\epsilon}{2} + \frac{\epsilon}{2} = \epsilon.
\end{aligned}
$$

Hence, if we let $r = T \circ R \circ f^{-1} \in \mathrm{emb}(f(K), T \circ R(K))$ then we obtain that $B_{K,K}(f, T \circ R) < \epsilon$. This holds for any $\epsilon$, and hence we have that $\mathcal{T} \circ \mathcal{R}$ has the MEP for $\mathcal{I}(\mathbb{R}^n, \mathbb{R}^m)$.

The proof in the case that $o \ge n$ follows with minor modification, and applying Lemma 7 point 1.

$\square$

### C.2.2 $S^1$ can not be mapped extendably to the trefoil knot

We first show that there are no maps $E := T \circ R$ where $R : \mathbb{R}^2 \to \mathbb{R}^3$ such that $T$ is a homeomorphism and $E(S^1)$ is a trefoil knot. We use the fact that the trivial knot $S^1$ and the trefoil knot $\mathcal{M} = f(S^1)$ are not equivalent, that is, there are no homeomorphisms in $\mathbb{R}^3$ that map $S^1$ to $\mathcal{M}$. Indeed, by (Murasugi, 2008, Section 3.2), the trefoil knot $\mathcal{M}$ and its mirror image are not equivalent, whereas the trivial knot $S^1$ and its mirror image are equivalent. Hence, $\mathcal{M}$ and $R(S^1)$ are not equivalent knots in $\mathbb{R}^3$. Thus by (Murasugi, 2008, Definition 1.3.1 and Theorem 1.3.1), we see that there is no orientation preserving homeomorphism $T : \mathbb{R}^3 \to \mathbb{R}^3$ such that $T(\mathbb{R}^3 \setminus R(S^1)) = \mathbb{R}^3 \setminus \mathcal{M}$. As the orientation of the map $T$ can be changed by composing $T$ with the reflection $J : \mathbb{R}^3 \to \mathbb{R}^3$ across the plane $\mathrm{Range}(R)$ that defines a homeomorphim $J : \mathbb{R}^3 \setminus R(S^1) \to \mathbb{R}^3 \setminus R(S^1)$, we see that there is no homeomorphism $T : \mathbb{R}^3 \to \mathbb{R}^3$ such that $T(\mathbb{R}^3 \setminus R(S^1)) = \mathbb{R}^3 \setminus \mathcal{M}$.

This example shows that the composition $E = T \circ R$ of a linear map $R$ and a coupling flow $T$ cannot have the property that $E(S^1) = f(S^1)$ for this embedding $f$. Moreover, the complement $\mathbb{R}^3 \setminus E(S^1)$ is never homeomorphic to $\mathbb{R}^3 \setminus f(S^1)$ for any such map $E$.

We now construct another example, similar to Figure 2, where an annulus that is mapped to a knotted ribbon in $\mathbb{R}^3$. To do this, replace the circle $S^1$ by an annulus $K = \{x \in \mathbb{R}^2 : 1/2 \le |x| \le 3/2\}$, that in the polar coordinates is $\{(r, \theta) : 1/2 \le r \le 3/2\}$ and define a map $F : K \to \mathbb{R}^3$ by defining in the polar coordinates

$$F(r, \theta) = f(\theta) + a(r - 1)v(\theta)$$

where $f : S^1 \to \Sigma_1 \subset \mathbb{R}^3$ is an smooth embedding of $S^1$ to a trefoil knot $\Sigma_1$ and $v(\theta) \in \mathbb{R}^3$ is a unit vector normal to $\Sigma_1$ at the point $f(\theta)$ such that $v(\theta)$ is a smooth function of $\theta$, and $a > 0$ is

a small number. In this case, $M_1 = F(K)$ is a 2-dimensional submanifold of $\mathbb{R}^3$ with boundary, which can visualizes $M_1$ as a knotted ribbon.

We now show that there are no maps $E = T \circ R$ such that $E(K) = F(K)$ where $T \colon \mathbb{R}^3 \to \mathbb{R}^3$ is an embedding, and $R \colon \mathbb{R}^2 \to \mathbb{R}^3$ injective and linear. The key insight is that if such a $T$ existed, then this implies that the trefoil knot is equivalent to $S^1$ in $\mathbb{R}^3$, which is known to be false.

Let $U_\rho(A)$ denote the $\rho$-neighborhood of the set $A$ in $\mathbb{R}^3$. It is easy to see that $\mathbb{R}^2 \setminus (\{0\} \times [-1, 1])$ is homeomorphic to $\mathbb{R}^2 \setminus \overline{B}_{\mathbb{R}^2}(0, 1)$, which is further homeomorphic to $\mathbb{R}^2 \setminus \{0\}$. Thus, using tubular coordinates near $\Sigma_1$ and a sufficiently small $\rho > 0$, we see that $\mathbb{R}^3 \setminus M_1$ is homeomorphic to $\mathbb{R}^3 \setminus U_\rho(\Sigma_1)$, which is further homeomorphic to $\mathbb{R}^3 \setminus \Sigma_1$. Also, when $R \colon \mathbb{R}^2 \to \mathbb{R}^3$ is an injective linear map, we see that $M_2 = R(K)$ is a un-knotted band in $\mathbb{R}^3$ and $\mathbb{R}^3 \setminus M_2$ is homeomorphic to $\mathbb{R}^3 \setminus \Sigma_2$. If $\mathbb{R}^3 \setminus M_1$ and $\mathbb{R}^3 \setminus M_2$ would be homeomorphic, then also $\mathbb{R}^3 \setminus \Sigma_1$ and $\mathbb{R}^3 \setminus \Sigma_2$ would be homeomorphic that is not possible by knot theory, see (Murasugi, 2008, Definition 1.3.1 and Theorem 1.3.1). This shows that there are no injective linear maps $R : \mathbb{R}^2 \to \mathbb{R}^3$ and homeomorphisms $\Phi : \mathbb{R}^3 \to \mathbb{R}^3$ such that $(\Phi \circ R)(K) = M_1$.

Similar examples can be obtained in a higher dimensional case by using a knotted torus Séquin (2011)[4] and their Cartesian products.

### C.2.3   The proof of Example 1

*Proof.*     (i) From (Puthawala et al., 2020, Theorem 15) we have that $\mathcal{R}^{o,m}$ can approximate any continuous function $f \in \mathrm{emb}(\mathbb{R}^n, \mathbb{R}^m)$. Further, clearly (T1) and (T2) both contain the identity map, thus Lemma 1 (i) applies.

(ii) Let $\mathcal{T}^m$ be the family autoregressive flows with sigmoidal activations defined in (Huang et al., 2018). By (Teshima et al., 2020, App. G, Theorem 1 and Proposition 7), $\mathcal{T}^m$ are sup-universal approximators in the space $\mathrm{Diff}^2(\mathbb{R}^m, \mathbb{R}^m)$ of $C^2$-smooth diffeomorphisms $\Phi : \mathbb{R}^m \to \mathbb{R}^m$. When $\mathcal{R}^{o,m}$ is one of (R1) or (R2) the network is always linear, hence the conditions are satisfied. If $\mathcal{R}^{o,m}$ is (R4), then $\mathcal{R}^{o,m}$ contains linear mappings, and if (R3), then we can shift the origin, so that $R(x)$ is linear on $K$. In all cases, Lemma 1 part (ii) applies.

$\square$

### C.2.4   The proof of Lemma 2

Given an $f \in \mathrm{emb}^\infty(K, \mathbb{R}^m)$ we first show that for $m \geq 2n+1$ there are always a diffeomorphisms $\Psi \colon \mathbb{R}^m \to \mathbb{R}^m$ so that $\Psi \circ f \colon \mathbb{R}^n \to \{0\}^n \times \mathbb{R}^{m-n}$. The existence of such a $\Psi$ borrows ideas from Whitney's embedding theorem (Hirsch, 2012, Theorems 3.4 & 3.5) and is constructed by iteratively constructing an injective projection.

Next if $m - n \geq 2n+1$, then we can apply (Madsen et al., 1997, Lemma 7.6), a consequence of the Tietze extension theorem, to show that $\Psi \colon \mathcal{M} \to \{0\}^n \times \mathbb{R}^{m-n}$ can be extended to a diffeomorphism on the entire space, $h \colon \mathbb{R}^m \to \mathbb{R}^m$. Hence $f(x) = \Psi^{-1} \circ h \circ R(x)$ for diffeomorphism $\Psi^{-1} \circ h \colon \mathbb{R}^m \to \mathbb{R}^m$ and zero-padding operator $R \colon \mathbb{R}^n \to \mathbb{R}^m$, and thus $f \in \mathcal{I}^\infty(K, \mathbb{R}^m)$.

This fact that for $m$ sufficiently large compared to $n$ such a diffeomorphism can always be extended is related to the fact that in 4-dimensions, all knots can be opened. This can be contrasted with the case in Figure 2.

Now we present our proof.

*Proof.* Let use next prove Eq. 9 when $m \geq 3n + 1$. Let

$$f \in \mathrm{emb}^k(\mathbb{R}^n, \mathbb{R}^m) \tag{35}$$

be a $C^k$ map and $\mathcal{M} = f(\mathbb{R}^n)$ be an embedded submanifold of $\mathbb{R}^m$.

---

[4]On the knotted torus, see `http://gallery.bridgesmathart.org/exhibitions/2011-bridges-conference/sequin`.

We have that $m \geq 3n + 1 > 2n + 1$. Let $S^{m-1}$ be the unit sphere of $\mathbb{R}^m$ and let

$$S\mathbb{R}^m = \{(x, v) \in \mathbb{R}^m \times \mathbb{R}^m : \|v\| = 1\}$$

be the sphere bundle of $\mathbb{R}^m$ that is a manifold of dimension $2m - 1$. By the proof's of Whitney's embedding theorem, by Hirsch, (Hirsch, 2012, Chapter 1, Theorems 3.4 and 3.5), there is a set of 'problem points' $H_1 \subset S^{m-1}$ of Hausdorff dimension $2n$ such that for all $w \in \mathbb{R}^m \setminus H_1$ the orthogonal projection

$$P_w : \mathbb{R}^m \to \{w\}^\perp = \{y \in \mathbb{R}^m : y \perp w\}$$

has a restriction $P_w|_{\mathcal{M}}$ on $\mathcal{M}$ defines an injective map

$$P_w|_{\mathcal{M}} : \mathcal{M} \to \{w\}^\perp.$$

Moreover, let $T_x \mathcal{M}$ be the tangent space of manifold $\mathcal{M}$ at the point $x$ and let us define another set of 'problem points' as

$$H_2 = \{v \in S^{m-1} : \exists x \in \mathcal{M}, v \in T_x \mathcal{M}\}.$$

For $w \in S^{m-1} \setminus H_2$ the map

$$P_w|_{\mathcal{M}} : \mathcal{M} \to \{w\}^\perp \subset \mathbb{R}^m$$

is an immersion, that is, it has an injective differential. The sphere tangent bundle $S\mathcal{M}$ of $\mathcal{M}$ has dimension $2n - 1$, and the set $H_2$ has the Hausdorff dimension at most $2n - 1$. Thus $H = H_1 \cup H_2$ has Hausdorff dimension at most $2n < m - 1$ and hence the set $S^{m-1} \setminus H$ is non-empty. For $w \in S^{m-1} \setminus H$ the map $P_w|_{\mathcal{M}} : \mathcal{M} \to \{w\}^\perp$ is a $C^k$ injective immersion and thus

$$\tilde{N} = P_w(\mathcal{M}) \subset \{w\}^\perp$$

is a $C^k$ submanifold.

Let $Z : P_w(\mathcal{M}) \to \mathcal{M}$ be the $C^k$ function defined by

$$Z(y) \in \mathcal{M}, \quad P_w(Z(y)) = y,$$

that is it is the inverse of $P_w|_{\mathcal{M}} : \mathcal{M} \to P_w(\mathcal{M})$, where $P_w(\mathcal{M}) \subset \{w\}^\perp$. Let $g : \tilde{N} = P_w(\mathcal{M}) \to \mathbb{R}$ be the function

$$g(y) = (Z(y) - y) \cdot w, \quad y \in P_w(\mathcal{M}).$$

Then $\tilde{N}$ is a $n$-dimensional $C^k$ submanifold of $(m-1)$-dimensional Euclidean space $H = \{w\}^\perp$ and $g$ is a $C^k$ function defined on it. By definition of a $C^k$ submanifold of $H$, any point $x \in \tilde{N}$ has a neighborhood $U \subset H$ with local $C^k$ coordinates $\psi : U \to \mathbb{R}^m$ such that $\psi(\tilde{N} \cap U) = (\{0\}^{m-1-n} \times \mathbb{R}^n) \cap \psi(U)$. Using these coordinates, we see that $g$ can be extended to a $C^k$ function in $U$. Using a suitable partition of unity, we see that there is a $C^k$ map $G : \{w\}^\perp \to \mathbb{R}$ that a $C^k$ extension of $g$ that is, $G|_{\tilde{N}} = g$.

Then the map

$$\Phi_1 : \mathbb{R}^m \to \mathbb{R}^m, \quad \Phi_1(x) = x - G(P_w(x))w$$

is a $C^k$ diffeomorphism of $\mathbb{R}^m$ that maps $\mathcal{M}$ to $m - 1$ dimensional space $\{w\}^\perp$, that is

$$\Phi_1(\mathcal{M}) \subset \{w\}^\perp.$$

In the case when $m \geq 3n + 1$, we can repeat this construction $n$ times. This is possible as $m - n \geq 2n + 1$. Then we obtain $C^k$ diffeomorphisms $\Phi_j : \mathbb{R}^m \to \mathbb{R}^m$, $j = 1, \ldots, n$ such that their composition $\Phi_n \circ \cdots \circ \Phi_1 : \mathbb{R}^m \to \mathbb{R}^m$ is a $C^k$-diffeomorphism such that which

$$\Phi_n \circ \cdots \circ \Phi_1(\mathcal{M}) \subset Y',$$

where $Y' \subset \mathbb{R}^m$ is a $m - n$ dimensional linear space. By letting $\Psi = Q \circ \Phi_n \circ \cdots \circ \Phi_1$ for rotation matrix $Q \in \mathbb{R}^{m \times m}$, we have that $Y := Q(Y') = \{0\}^n \times \mathbb{R}^{m-n}$. Also, letting $\phi : X = \mathbb{R}^n \times \{0\}^{m-n} \to \mathbb{R}^m$ be the map

$$\phi(x, 0) = \Psi(f(x)) \in Y,$$

where $f$ is the function given in Eq. 35, we have that $\phi : X \to Y$ is a $C^k$-diffeomorphism. We observe that $m - n \geq 2n + 1$ and so we can apply (Madsen et al., 1997, Lemma 7.6) to extend $\phi$ to a $C^k$-diffeomorphism

$$h : \mathbb{R}^m \to \mathbb{R}^m$$

such that $h|_X = \phi$. Note that (Madsen et al., 1997, Lemma 7.6) concerns an extension of a homeomorphism, but as the extension $h$ is given by an explicit formula which is locally a finite sum of $C^k$ functions, the same proof gives a $C^k$-diffeomorphic extension $h$ to a diffeomorphism $\phi$. This technique is called the 'clean trick'.

Finally, to obtain the claim, we observe that when $R : \mathbb{R}^n \to \mathbb{R}^m$, $R(x) = (x, 0) \in \{0\}^n \times \mathbb{R}^{m-n}$ is the zero padding operator, we have

$$f(x) = \Psi^{-1}(\phi(R(x))), \quad x \in \mathbb{R}^n.$$

As $h|_X = \phi$ and $R(x) \in X$, this yields

$$f(x) = \Psi^{-1}(h(R(x))), \quad x \in \mathbb{R}^n,$$

that is,

$$f = E \circ R$$

where $E = \Psi^{-1} \circ h : \mathbb{R}^m \to \mathbb{R}^m$ is a $C^k$ diffeomorphism. Thus $f \in \mathcal{I}^k(\mathbb{R}^n, \mathbb{R}^m)$. This proves Eq. 9 when $m \geq 3n + 1$. $\square$

### C.2.5 THE PROOF OF LEMMA 3

*The proof of Lemma 3.* Let $f = F_2 \circ F_1$ where $F_2 \in \mathcal{F}^{o,m}$ and $F_1 \in \mathcal{F}^{n,o}$ and $\epsilon > 0$ be given, and let $E^{o,m}$. Clearly, $B_{K,W}(f, E) \leq B_{K,W}(F_2, E)$ and so by the $m, o, o$ MEP of $\mathcal{E}^{o,m}$ with respect to $\mathcal{F}^{o,m}$, we have the existence of an $r_m \in \text{emb}(f(K), E^{o,m})$ such that $\|I - r\|_{L^\infty(f(K))} < \epsilon$. $K_o := (E^{o,m})^{-1} \circ r \circ f(K)$ is compact, hence $E^{o,m}$ is Lipschitz on $K_o$, so we can apply Lemma 7 point 4, so

$$B_{K,W}(f, E^{o,m} \circ E^{p,o}) \leq \|I - r\|_{L^\infty(f(K))} + \text{Lip}(E^{o,m}) B_{K,W}((E^{o,m})^{-1} \circ r \circ f, E^{p,o}). \quad (36)$$

But, because $f \in \mathcal{F}^{o,m} \circ \mathcal{F}^{n,o}$, we can choose a $E^{p,o} \in \mathcal{E}_1^{p,o}$ so that $B_{K,W}((E^{o,m})^{-1} \circ r \circ f, E^{p,o}) \leq \frac{\epsilon}{2} \text{Lip}(E^{o,m})$ which, combined with Eqn. 36, proves the result. $\square$

### C.2.6 THE PROOF OF LEMMA 4

*The proof of Lemma 4.* Recall that $\mathcal{F} \subset \text{emb}(\mathbb{R}^n, \mathbb{R}^m)$. Suppose that $\mathcal{E}_2^{o,m}$ does not have the $m, n, o$ MEP with respect to $\mathcal{F}$, then there are some $\epsilon > 0$ and $f \in \mathcal{F}$ so that

$$\forall E^{o,m} \in \mathcal{E}_2^{o,m} \, \forall W_1 \subset\subset \mathbb{R}^o, \quad B_{K,W_1}(f, E_2^{o,m}) \geq \epsilon. \quad (37)$$

From Lemma 7 point 2, we have that

$$\epsilon \leq B_{K,W_1}(f, E_2^{o,m}) \leq B_{K,W}(f, E_2^{o,m} \circ E_1^{p,o}) \quad (38)$$

for all $E_1^{p,o} \in \mathcal{E}_1^{p,o}$ and for all compact sets $W \subset \mathbb{R}^p$ that satisfy $E_1^{p,o}(W_1) \subset W$. We observe that if $W' \subset \mathbb{R}^p$ is a compact set such that $W' \subset W$, we have

$$B_{K,W}(f, E_2^{o,m} \circ E_1^{p,o}) \leq B_{K,W'}(f, E_2^{o,m} \circ E_1^{p,o})$$

Thus, inequality Eq. 38 holds for all $E_1^{p,o} \in \mathcal{E}_1^{p,o}$ and for all compact sets $W \subset \mathbb{R}^p$. Summarising, we have seen that there are $f \in \mathcal{F}$ and $\epsilon > 0$ such that for all $E_1^{p,o} \in \mathcal{E}_1^{p,o}$ and for all compact sets $W \subset \mathbb{R}^p$ we have $\epsilon \leq B_{K,W}(f, E_2^{o,m} \circ E_1^{p,o})$. Hence $\mathcal{E}_2^{o,m}$ does not have the $m, n, o$ MEP with respect to $\mathcal{F}$, and we have obtained a contradiction, which proves the result. $\square$

## C.3 UNIVERSALITY

### C.3.1 THE PROOF OF THEOREM 1

*The proof of Theorem 1.* First we prove the claim under the assumptions (i).

First we prove the claim under assumption (i).

Let $W \subset \mathbb{R}^n$ be an open relatively compact set. From Lemma 3 we have that

$$\mathcal{E}^{n,m} := \mathcal{E}_L^{n_{L-1},m} \circ \cdots \circ \mathcal{E}_1^{n,n_1} \quad (39)$$

has the $m, n, n$ MEP w.r.t. $\mathcal{F} := \mathcal{F}_L^{n_{L-1}, m} \circ \cdots \circ \mathcal{F}_1^{n, n_1}$. Thus for any $\epsilon_1 > 0$, we have an $\tilde{E} \in \mathcal{E}^{n,m}$ s.t. $B_{K,W}(F, \tilde{E}) < \epsilon_1$.

From Lemma 7 point 5, we have the existence of a $\mu' \in \mathcal{P}(W)$ so that $W_2\left(F_{\#}\mu, \tilde{E}_{\#}\mu'\right) < \epsilon_1$. By convolving $\mu'$ with a suitable mollifier $\phi$, we can obtain a measure $\mu'' = \mu' * \phi \in \mathcal{P}(W)$ that is absolutely continuous with respect to the Lebesgue measure so that

$$W_2\left(\mu', \mu''\right) < \frac{\epsilon_1}{1 + \operatorname{Lip}(\tilde{E})},$$

see (Ambrosio et al., 2008, Lemma 7.1.10.), and so $W_2\left(\tilde{E}_{\#}\mu', \tilde{E}_{\#}\mu''\right) < \epsilon_1$. Hence,

$$W_2\left(F_{\#}\mu, \tilde{E}_{\#}\mu''\right) < 2\epsilon_1. \tag{40}$$

Next, from universality of $\mathcal{T}_0^n$ for any $\epsilon_2 > 0$, we have the existence of a $T_0 \in \mathcal{T}_0^n$ so that $W_2\left(\mu'', T_{0\#}\mu\right) < \epsilon_2$. From Lemma 7 points 3 and 4 we have that

$$W_2\left(F_{\#}\mu, \tilde{E} \circ T_{0\#}\mu\right) \leq 2\epsilon_1 + \epsilon_2 \operatorname{Lip}(\tilde{E}). \tag{41}$$

For a given $\epsilon > 0$, choosing $\epsilon_1 < \frac{\epsilon}{4}$ and $\epsilon_2 < \frac{\epsilon}{2(1+\operatorname{Lip}(\tilde{E}))}$ yields that the map $E = \tilde{E} \circ T_0 \in \mathcal{E}$ is such that $W_2\left(F_{\#}\mu, E_{\#}\mu\right) < \epsilon$. This yields the result.

Next we prove the claim under the assumptions (ii). By our assumptions, in the weak topology of the space $C^2(\mathbb{R}^{n_j}, \mathbb{R}^{n_j})$, the closure of the set $\mathcal{T}^{n_j} \subset C^2(\mathbb{R}^{n_j}, \mathbb{R}^{n_j})$ contains the space of $\operatorname{Diff}^2(\mathbb{R}^{n_j}, \mathbb{R}^{n_j})$. Moreover, by our assumptions $\mathcal{R}^{n_{j-1}, n_j}$ contains a linear map $R$. We observe that as $\mathcal{R}^{n_{j-1}, n_j}$ is a space of expansive elements, the map $R$ is injective. and hecne by Lemma 1, the family

$$\mathcal{E}_j^{n_{j-1}, n_j} = \mathcal{T}^{n_j} \circ \mathcal{R}^{n_{j-1}, n_j}$$

has the MEP w.r.t. $\mathcal{F} = \mathcal{I}^1(\mathbb{R}^n, \mathbb{R}^m)$. By Lemma 2, we have that $\mathcal{I}^1(\mathbb{R}^n, \mathbb{R}^m)$ coincides with the space $\operatorname{emb}^1(\mathbb{R}^n, \mathbb{R}^m)$. Finally, by the assumption that $\mathcal{T}_0^{n_0}$ is dense in the space of $C^2$-diffeomorphism $\operatorname{Diff}^2(\mathbb{R}^{n_\ell})$ implies that $\mathcal{T}_0^{n_0}$ is a $L^p$-universal approximator for the set of $C^\infty$-smooth triangular maps for all $p < \infty$. Hence by Lemma 3 in Appendix A of Teshima et al. (2020), $\mathcal{T}_0^{n_0}$ is a distributionally universal. From these the claim in the case (ii) follows in the same way as the case (i) using the family $\mathcal{F} = \operatorname{emb}^1(\mathbb{R}^n, \mathbb{R}^m)$. $\qquad \square$

### C.3.2 THE PROOF OF LEMMA 5

*The proof of Lemma 5.* The proof follows from taking the logical negation of the MEP for $\mathcal{F}$. If the MEP is not satisfied, then there is some $f \in \mathcal{F}$ so that $B_{K,W}(f, E)$ is never smaller than $\epsilon > 0$ for all $E \in \mathcal{E}$. Applying the definition of $B_{K,W}(f, E)$ from Eqn. 6 yields the result. $\qquad \square$

### C.3.3 THE PROOF OF COR. 1

*The proof of Cor. 1.* The proof of Eqn 12 follows from the definition of the MEP.

The proof of Eqn. 13 follows from applying Lemma 7 point 4, and applying the same arguments as in the proof of Theorem 1 so that $\lim_{i \to \infty} \frac{1}{\operatorname{Lip}(E_i)} B_{K,W}(E_i^{-1} \circ r \circ f, E_i') \to 0$, and so that $W_2\left(F_{\#}\mu, E_i \circ E_{i\#}'\mu'\right) = \epsilon_i$ where $\epsilon_i \to 0$ as $i \to \infty$.. From the distributional universality of $\mathcal{T}_0^n$, and continuity of $E_i \circ E_i'$ we have the existence of a $T_i \in \mathcal{T}_0^n$ so that $W_2\left(F_{\#}\mu, E_i \circ E_i' \circ T_{i\#}\mu\right) = 2\epsilon_i$. Choosing $\epsilon_i$ so that $\lim_{i \to \infty} \epsilon_i$ yields the result. $\qquad \square$

## APPENDIX D  ADDITIONAL PROPERTIES

### D.1  LAYER-WISE PROJECTION

Here we provide the details of our closed-form layerwise projection algorithm The flow layers are injective, and are often implemented to be numerically easy to invert. Thus, the crux of the algorithm

comes from inverting the injective expansive layers, $R$. The range of the ReLU layer is piece-wise affine, hence the inversion follows a two-step program. First, identify which affine piece (described algebraically, onto which sign pattern) to project. Second, project to this point using a standard least-squares solver.

The second step is always straight-forward to analyze, but the first is more complicated.

The key step in our algorithm is the fact that for the specific choice of weight matrix $W = \begin{bmatrix} B \\ -DB \end{bmatrix}$, given any $y \in \mathbb{R}^{2n}$, we can always solve the least-squares inversion problem exactly.

We prove this result in several parts given below.

1. For any $y \in \mathbb{R}^{2n}$, $M_y W \in \mathbb{R}^{n \times n}$ is full-rank.

2. If $[y]_i \neq [y]_{i+n}$ for each $i = 1, \ldots, n$, then the $\mathrm{argmin}$ in Eqn. 17 is well defined, i.e. that there is a unique minimizer. Otherwise there are $2^I$ minimizers, where $I$ is the number of distinct $i$ such that $[y]_i = [y]_{i+n}$.

3. If $\tilde{M}_y = \begin{bmatrix} \Delta_y & (I^{n \times n} - \Delta_y) \end{bmatrix}$, then

$$\min_{x \in \mathbb{R}^n} \|y - R(x)\|_2^2 = \min_{x \in \mathbb{R}^n} \|M_y(y - Wx)\|_2^2 + \left\| \tilde{M}_y y \right\|_2^2. \qquad (42)$$

4. We verify Eqn. 17.

*The proof of Theorem 2.*  1. Using the definition of $M_y$, we have,

$$M_y \begin{bmatrix} B \\ -DB \end{bmatrix} = (I^{n \times n} - \Delta_y) B - \Delta_y DB = (I^{n \times n} - \Delta_y - \Delta_y D) B. \qquad (43)$$

But, $(I^{n \times n} - \Delta_y - \Delta_y D)$ is a full-rank diagonal matrix (with entries either $1$ or $[D]_{i,i}$), and $B$ is full rank by assumption, hence $M_y \begin{bmatrix} B \\ -DB \end{bmatrix}$ is too.

2. Because $B$ is square and full rank there exists a basis[5] $\left\{ \hat{b}_i \right\}_{i=1,\ldots,n}$ of $\mathbb{R}^n$ such that

$$\left\langle \hat{b}_j, b_i \right\rangle = \begin{cases} 1 & \text{if } i = j \\ 0 & \text{if } i \neq j \end{cases}. \qquad (44)$$

For an $x \in \mathbb{R}^n$, let $\alpha_i = \langle x, b_i \rangle$ for $i = 1, \ldots, n$ be the expansion of $x$ in the $\hat{b}_i$ basis.

$$\min_{x \in \mathbb{R}^n} \|y - R(x)\|_2^2 = \min_{x \in \mathbb{R}^n} \sum_{i=1}^{2n} [y - R(x)]_i^2 \qquad (45)$$

$$= \sum_{i=1}^{n} \min_{x_i \in \mathbb{R}} \left([y]_i - \max(\langle x, b_i \rangle, 0)\right)^2 + \left([y]_{i+n} - \max(\langle x, -[D]_{ii} b_i \rangle, 0)\right)^2 \qquad (46)$$

We now consider minizing Eqn. 46 by minimizing the basis expansion in terms of $\alpha_i$,

$$\sum_{i=1}^{n} \min_{\alpha_i \in \mathbb{R}} \left([y]_i - \max(\alpha_i, 0)\right)^2 + \left([y]_{i+n} - \max(-[D]_{ii} \alpha_i, 0)\right)^2 \qquad (47)$$

Eqn. 47 is clearly minimized by minizing each term in the sum, hence we search for a minimizer of the $i$'th term

$$\min_{\alpha_i \in \mathbb{R}} \left([y]_i - \max(\alpha_i, 0)\right)^2 + \left([y]_{i+n} - \max(-[D]_{ii} \alpha_i, 0)\right)^2 \qquad (48)$$

---

[5]Namely the columns of the matrix $B^{-1}$

Noting $f(\alpha_i)$ as the quantity inside the minimum of Eqn. 48, we consider the positive, negative and zero $\alpha_i$ cases of Eqn. 48 separately and we get

$$\min_{\alpha_i \in \mathbb{R}^+} f(\alpha_i) = \min_{\alpha_i \in \mathbb{R}^+} \left([y]_i - \alpha_i\right)^2 + [y]_{i+n}^2 = [y]_{i+n}^2 \tag{49}$$

$$\min_{\alpha_i \in \mathbb{R}^-} f(\alpha_i) = \min_{\alpha_i \in \mathbb{R}^+} [y]_i^2 + \left([y]_{i+n} + [D]_{ii}\, \alpha_i\right)^2 = [y]_i^2 \tag{50}$$

$$f(0) = [y]_i^2 + [y]_{i+n}^2. \tag{51}$$

If $[y]_{i+n} > [y]_i$, then the minimizer of Eqn. 48 is $\alpha_i = -\frac{[y]_{i+n}^2}{[D]_{ii}} < 0$. Conversely if $[y]_{i+n} < [y]_i$ then the minimzer of Eqn. 48 is $\alpha_i = [y]_i > 0$. This argument applies all $i = 1, \ldots, n$, and hence if $[y]_i \neq [y]_{i+1}$ for all $i = 1, \ldots, n$ then the minimizing $x$ is unique.

If $[y]_i = [y]_{i+1}$ then there are exactly two minimizers of $f(\alpha_i)$, $-\frac{[y]_{i+n}^2}{[D]_{ii}}$ and $[y]_i$, for both of which $f(\alpha_i) = [y]_i^2 = [y]_{i+n}^2$.

3. If we suppose that $[y]_{i+n} - [y]_i > 0$, then $[c(y)]_i = 0$ and $[c(y)]_{i+n} > 0$, thus $[\Delta_y]_{ii} = 1$, hence if we let $x_{\min}$ be the minimizing $x$ from part 1, then

$$\left([y]_i - \max(\langle x_{\min}, b_i \rangle, 0)\right)^2 + \left([y]_{i+n} - \max(\langle x_{\min}, -[D]_{ii}\, b_i \rangle, 0)\right)^2 \tag{52}$$

$$= [y]_i^2 + \left([y]_{i+n} - \max(\langle x_{\min}, -[D]_{ii}\, b_i \rangle, 0)\right)^2 \tag{53}$$

$$= \left[\tilde{M}_y y\right]_i^2 + [M_y\, (y - W x_{\min})]_i^2 \tag{54}$$

If $[y]_{i+n} - [y]_i \leq 0$ then we have

$$\left([y]_i - \max(\langle x_{\min}, b_i \rangle, 0)\right)^2 + \left([y]_{i+n} - \max(\langle x_{\min}, -[D]_{ii}\, b_i \rangle, 0)\right)^2 \tag{55}$$

$$= \left([y]_i - \max(\langle x_{\min}, b_i \rangle, 0)\right)^2 + [y]_{i+n}^2 \tag{56}$$

$$= [M_y\, (y - W x_{\min})]_i^2 + \left[\tilde{M}_y y\right]_i^2. \tag{57}$$

Thus combining Eqn.s 45, 46, 54 and 57 for each $i = 1, \ldots, n$, we have that

$$\min_{x \in \mathbb{R}^n} \|y - R(x)\|_2^2 = \min_{x \in \mathbb{R}^n} \|M_y\, (y - Wx)\|_2^2 + \|M_y y\|_2^2. \tag{58}$$

4. For the final point, combining all of the above points we have

$$\min_{x \in \mathbb{R}^n} \|y - R(x)\|_2^2 = \min_{x \in \mathbb{R}^n} \|M_y\, (y - Wx)\|_2^2. \tag{59}$$

Further we have from Point 1 that $M_y W$ is full rank, hence $(M_y W)^{-1} M_y y = R^\dagger(y)$ is a minimizer of Eqn. 59. If $[y]_i \neq [y]_{i+n}$ for all $i = 1, \ldots, n$ then Part 2 applies, and $R^\dagger(y)$ is the unique minimizer of $\|y - R(x)\|_2^2$. In either case, we have that $R^\dagger(y)$ is a minimizer.

$\square$

## D.2 Bayesian Uncertainty Quantification

We now discuss the process for making the network amenable to Bayesian Uncertainty Quantification (Bayesian UQ).

**Assumption 2.** *Let $y_0 \in \mathbb{R}^n$ be given, let $y_1 = T_0 y_0$ and for each $\ell = 2, \ldots, L$, then let $y_\ell := R_\ell(y_{\ell-1/2})$ and $y_{\ell-1/2} := T_{\ell-1}(y_{\ell-1})$. We assume that for $\ell = 1, \ldots, L-1$, $T_\ell(y_\ell)$, $R_\ell(y_{\ell-1/2})$ and $T_0(y_0)$ are differentiable.*

**Lemma 8.** *If a network $\mathcal{F}$ of the form of Eqn. 1 satisfies Assumption 2, then we have the following upper bound on the log-likelihood,*

$$\log p_{\mathbf{y}}(y_0) \leq \log p_{\mathbf{z}}(F^{-1}(y_0)) + \frac{1}{2} \sum_{\ell=1}^{L} \log \left| \det \frac{\partial R_\ell}{\partial y_{\ell-1/2}}^T (y_{\ell-1/2}) \frac{\partial R_\ell}{\partial y_{\ell-1/2}} (y_{\ell-1/2}) \right|$$
$$+ \sum_{\ell=0}^{L} \log \left| \det \frac{\partial T_\ell}{\partial y_\ell} (y_\ell) \right|. \tag{60}$$

*Proof.* The proof of Lemma 8 follows from computing the log of,

$$p_{\mathbf{y}}(y_0) = p_{\mathbf{z}}(F^{-1}(y_0)) \left| \det \frac{\partial F^{-1}}{\partial y_0}\bigg|_{F(\mathbb{R}^n)} \right| \tag{61}$$

and then applying definition 1 and expanding term using the laws of logarithms we obtain.

$$\log p_{\mathbf{y}}(y) \leq \log p_{\mathbf{z}}(F^{-1}(y)) + \frac{1}{2} \sum_{\ell=1}^{L} \log \underbrace{\left| \det \frac{\partial R_\ell}{\partial y_{\ell-1/2}}^T (y_{\ell-1/2}) \frac{\partial R_\ell}{\partial y_{\ell-1/2}} (y_{\ell-1/2}) \right|}_{\boxed{R}}$$
$$+ \sum_{\ell=0}^{L} \log \underbrace{\left| \det \frac{\partial T_\ell}{\partial y_\ell} (y_\ell) \right|}_{\boxed{T}}. \tag{62}$$

for $\ell = 1, \ldots, L$, see (Kothari et al., 2021, Appendix B) for details. $\qquad\square$

We note here that in Equation 60 we have an inequality rather than an equality. The inequality comes from the expansive layers, as the flow-layers are all endomorphisms, i.e. $\mathbb{R}^{n_\ell} \to \mathbb{R}^{n_\ell}$. We now remark on the ease of approximating, or at least bounding, $\boxed{T}$ and $\boxed{R}$ in Eqn. 62.

Values for $\boxed{R}$ when $\mathcal{R}$ is (R1) or (R2) are straightforward to compute or approximate[6]. When $\mathcal{R}^{n,m}$ is (R3), $R \in \mathcal{R}$ are not differentiable, but are differentiable a.e.. Further, $\nabla R_\ell(x_{n_\ell})$ can a.e. be computed by applying a simple mask 0 and 1 mask to the weight matrices for an exact computation of $\boxed{R}$. Additionally, from Hutchinson (1989); Kothari et al. (2021) we have the following formula that approximates $\boxed{R}$,

$$\log \left| \det \frac{\partial R_\ell}{\partial y_\ell}^T \frac{\partial R_\ell}{\partial y_\ell} \right| = -\operatorname{Tr} \left( \frac{1}{k} \sum_{k=1}^{\infty} \left( I - \beta \frac{\partial R_\ell}{\partial y_\ell}^T \frac{\partial R_\ell}{\partial y_\ell} \right)^k \right) - d \log \beta$$
$$\approx -\mathbb{E}_\nu \left[ \frac{1}{k} \sum_{k=1}^{\infty} \nu^T \left( I - \beta \frac{\partial R_\ell}{\partial y_\ell}^T \frac{\partial R_\ell}{\partial y_\ell} \right)^k \nu \right] - d \log \beta. \tag{63}$$

where $\eta \in \mathbb{R}^{n_\ell}$. Computation of the $\boxed{T}$ term is often easier, comparatively. This is because these networks are often designed to make this value as easy to compute as possible. For example the NICE Dinh et al. (2014), Real-NVP Dinh et al. (2016), and GLOW Kingma & Dhariwal (2018) architectures are all designed so that $\boxed{T}$ can be computed form a short scalar product.

### D.3    BLACK-BOX RECOVERY

We now discuss assumptions that enable black-box recovery of the weights of our entire network post-training.

**Assumption 3.** *For each $\ell = 1, \ldots, L$, $\mathcal{R}_\ell$ is an affine* ReLU *layer. Each $\mathcal{T}_\ell$ and $\mathcal{T}_0$ is constructed from a finite number of affine* ReLU *layers.*

---

[6]for example from a *LU* decomposition, see (Golub, 1996, Section 3.2)

*Remark* 3. If a network $\mathcal{F}$ of the form of Eqn. 1 satisfies Assumption 3, then given the range of the network, the range of the network can be recovered exactly.

Further, if the linear region assumption from Rolnick & Körding (2020) is satisfied, then the exact weights are recovered, subject to two natural isometries discussed below.

*Remark* 4. The ReLU part of Assumption 3 is for all examples in Sec. 2.1. Further it is also satisfied by both flows considered in Sec. 2.2, provided that the various $g_i$ are given by layers of affine ReLU's.

In Rolnick & Körding (2020), the authors show that, although ReLU networks depend on the value of their weight matrix in non-linear ways, it is still possible to recover the exact weights of a given ReLU network in a black-box way, subject to natural isometrics. The authors show that this is possible not only in theory, but in numerical applications as well.

The works of Rolnick & Körding (2020); Bui Thi Mai & Lampert (2020) imply that provided the activation functions of the expressive elements are ReLU then the entire network can be recovered in a black-box way. Further, provided that either the 'linear region assumption' from Rolnick & Körding (2020) or the generality assumption from Bui Thi Mai & Lampert (2020) is satisfied, then the entire network can be recovered uniquely modulo the natural isometries of rescaling and permutation of weight matrices.

First we describe the two natural isometries of scaling and permutation. Consider the following function

$$f(x) = W_2 \phi(W_1 x) \tag{64}$$

where $\phi$ is coordinate-wise homogeneous degree 1 (such as ReLU) and $W_1 \in \mathbb{R}^{n_1 \times n_2}$ and $W_2 \in \mathbb{R}^{n_2 \times n_3}$. If we let $P \in \mathbb{R}^{n_2 \times n_2}$ be any permutation matrix, and $D_+$ be a diagonal matrix with strictly positive elements, then we can write

$$f(x) = W_2 P' D_+^{-1} \phi(D_+ P W_1 x) \tag{65}$$

as well. Thus ReLU networks can only ever be uniquely given subject to these two isometries. When describe unique recovery in the rest of this section, we mean modulo these two isometries.

In Rolnick & Körding (2020), the authors describe how all parameters of a ReLU network can be recovered uniquely (called reverse engineered in Rolnick & Körding (2020)), subject to the so called 'linear[7] region assumption', LRA.

The input space $\mathbb{R}^n$ can be partitioned into a finite number of open $\{\mathcal{S}_i\}_{i=1}^{n_i}$, where for each $k$, $F(x) = \mathbf{W}_k i + \mathbf{b}_i$, i.e. the network corresponds to an affine polyhedron in the output space. The algorithms (Rolnick & Körding, 2020, Alg.s 1 & 2) are roughly described below.

First, identify at least one point within each affine polyhedra $\{\mathcal{H}_j\}_{j=1}^{n_j}$. Then identify the boundaries between polyhedra. The boundaries between sections are always one affine 'piece' of piecewise hyperplanes $\{\mathcal{H}_j\}_{j=1}^{n_j}$. These $\{\mathcal{H}_j\}_{j=1}^{n_j}$ are the central objects which indicate the (de)activation of an element of a ReLU somewhere in the network. If the $\mathcal{H}_j$ are full hyperplanes, then the ReLU that is (de)activates occurs in the first layer of the network. If $\mathcal{H}_j$ is not a full hyperplane, then it necessarily has a bend where it intersects another hyperplane $\mathcal{H}_{j'}$. Further, except for a Lebesgue measure 0 set, when $\mathcal{H}_j$ intersects $\mathcal{H}_{j'}$ the latter does not have a bend. If this is the case, then $\mathcal{H}_{j'}$ corresponds to a ReLU (de)activation in an earlier layer than $\mathcal{H}_j$. In this way the activation functions of every layer can be deduced. Once this is done, the normals of the hyperplanes can be used to infer the row-vectors of the various weight matrices, letting one recover the entire network.

The above algorithm recovers all of the weights exactly provided that the LRA is satisfied. The LRA is satisfied if for every distinct $\mathcal{S}_i$ and $\mathcal{S}_{i'}$, either $\mathbf{W}_i \neq \mathbf{W}_{i'}$ or $\mathbf{b}_i \neq \mathbf{b}_{i'}$. That is, different sign patterns produce different affine sections in the output space. This is a natural assumption, as the algorithm as described above reconstruction works by first detecting the boundaries between adjacent affine polyhedra, which is only possible if the LRA holds.

Given the weights of a network there is currently no simple way to detect if the LRA is satisfied, to our knowledge. Nevertheless the authors of Rolnick & Körding (2020) show that if it is satisfied,

---

[7]The use of 'linear' in this context is somewhat non-standard, and instead means affine. In this section we use the term 'linear region assumption', but use 'affine' where Rolnick & Körding (2020) would use 'linear' to preserve mathematical meaning.

then unique recovery follows. Nevertheless recovery of the range of the entire network is possible, but this recovery may not be unique.

In Bui Thi Mai & Lampert (2020) the authors also consider the problem of recovering weights of a ReLU neural network, however the authors therein study the question of when there exist isometries beyond the two natural ones described above. In particular the main result (Bui Thi Mai & Lampert, 2020, Theorem 1) shows the following. Let $\mathcal{E}^{n_0, n_L}$ be a ReLU network that is $L$ layers deep and non-increasing. Suppose that $E_1, E_2 \in \mathcal{E}^{n_0, n_L}$, $E_1$ and $E_2$ are general[8] and for all $x \in \mathbb{R}^{n_0}$ $E_1(x) = E_2(x)$, then $E_1$ is parametrically identical to $E_2$ subject to the two natural isometries.

This work provides the stronger result, however does not apply to the networks that we consider out of the box. It does apply to our expressive elements (provided that they use ReLU activation functions, and are non-increasing), but not necessarily apply to the network on the whole.

---

[8]A set it in a topology is general if it's complement is closed and nowhere dense.

