# OpenReview forum: "Universal Joint Approximation of Manifolds and Densities by Simple Injective Flows"
_ICLR.cc/2022/Conference — ICLR 2022 Submitted_

### Official Review · Reviewer_WKWr · 2021-10-31

**Correctness:** 4
**Technical Novelty And Significance:** 4
**Empirical Novelty And Significance:** 4
**Recommendation:** 8
**Confidence:** 4

**Main Review:**

The paper is theoretical and mathematically fruitful. The embedding property discussed here is an important geometric property and should be useful in understanding the structure of neural networks. The write-up is a little bit too abstract and mathematical. It would be great if the authors can use a more understandable way to present their results in order that the work can reach a larger audience.

**Summary Of The Paper:**

This paper analyzes neural networks composed of bijective flows and injective expansive elements and showed that they are universal approximators of a large class of manifolds. A new mathematical notion called the embedding gap is proposed, which measures how far one continuous manifold is from embedding another.

**Summary Of The Review:**

(1) Good work. Mathematically profound and results may have potential applications;
(2) Presentation too mathematical and might reach a larger audience if it can be revised to be more understandable.

---

> ### Author Response · Authors · 2021-11-23
> **Thank you for the positive assessment + many clarifications in the updates manuscript**
>
> Thank you for the positive assessment. We are very pleased that you enjoyed the paper!
>
> *The paper is theoretical and mathematically fruitful. The embedding property discussed here is an important geometric property and should be useful in understanding the structure of neural networks. The write-up is a little bit too abstract and mathematical. It would be great if the authors can use a more understandable way to present their results in order that the work can reach a larger audience.*
>
> *(1) Good work. Mathematically profound and results may have potential applications; (2) Presentation too mathematical and might reach a larger audience if it can be revised to be more understandable.*
>
> We have invested considerable effort to make the narrative more understandable, for example by constructing concrete non-approximable manifold-supported distributions. We have also endeavored to emphasize with more clarity the nature of the topological difficulties. We hope that the description of the difficulties, combined with the new results for $m > 3n$ as well as the new examples, will give nonspecialists a good sense of the mathematical and practical difficulties put forth by injective flows, without the need to absorb all of the new notation.

---

### Official Review · Reviewer_4sjW · 2021-11-01

**Correctness:** 4
**Technical Novelty And Significance:** 4
**Empirical Novelty And Significance:** Not applicable
**Recommendation:** 6
**Confidence:** 4

**Main Review:**

## Strengths

* The paper makes good strides in the understanding of injective normalizing flows.

## Weaknesses

* The results feel weak. This is best exemplified by the class $\mathcal{I}(X, Y)$, which is a very strong assumption to make on our target embedding. This class is **very** close to the core construction in Brehmer and Cranmer 2020, so a result showing universal approximation does not feel particularly illuminating. In particular, since members of $\mathcal{I}(X, Y)$ are $\Phi \circ R$, where $R$ is affine and $\Phi$ is a normalizing flow, do the results not follow simply by approximating both? This feels rather straightforward. Furthermore, while the paper claims that it has to account for just topological obstructions, this class of functions limits for more than just that. For example, a map from $\mathbb{R}^n \to B_0^n(1) \subset \mathbb{R}^m$ is correct topologically but it is not considered in the proofs.
* The paper heavily invokes the universality and other properties of its constituent neural networks in the proofs. In particular, most of the proofs are simple applications of universality, follow directly from definitions, and/or rely on canonical tricks in the field (e.g. convolution with a mollifier). Along with the above observation, this lack of depth in the proof gives me the impression that not much real insight is being gained.
* The paper, as written, is very confusing. This is important as these issues significantly hamper the accessibility of the paper. I would suggest that the authors make a thorough pass through the paper to clear up these issues. I included some of the major ones, but there are certainly others.
    * There are many typos throughout, even in the abstract.
    * There are very poor notational conventions. For example, in Theorem 1 the $\mathcal{F}$ is overloaded.
    * The paper could be better structured. For example, the purpose for the embedding gap and MEP are not made clear; in particular, they are introduced with almost no motivation. Explaining why one has to leverage these constructions would help readers make sense of why they are introduced this way. Other sections are introduced rather strangely (e.g. the topological obstruction section is motivated before the section and leads directly into the definition of MEP without pause).
    * Most of the intermediate lemmas used to prove the theorem have little reason for being in the main paper. For example, Lemma 1 just proves basic properties of the definition and is unused except for in the appendix; it is not a main result and its purpose can not be understood unless one dives into the details of all proofs.
    * I was also confused about the previous work section. In particular, it seems very strange to me that the discussion about the core architectures that the paper prove are universal were moved to the appendix, while general inverse problem applications were given several paragraphs.
* The other results are very ancillary and oftentimes not explored/explained thoroughly. In particular, the last three results seem unrelated (and furthermore do not provide any more real insight). In particular, Section 3.4 (the results on layerwise projection) should be tested empirically. Appendix D.2 (on bayesian uncertainty quantification) is a straightforward application of standard log probability arguments. Finally, Appendix D.3 (the results on black box estimation) seems incomplete. There is no proof for Lemma 8, and I found the last paragraph, which is also one of the few that talks about the current work rather than Rolnick & Kording (2020), to be very confusing.

## Small Stuff

* The links to the lemmas are broken.

**Summary Of The Paper:**

This paper analyzes some theoretical properties of injective normalizing flows. In particular, it focuses on universal approximation and (to a lesser extent), layer-specific projections, bayesian uncertainty quantification, and black-box recovery of network weights.

**Summary Of The Review:**

While I believe the core idea and motivation are strong, I found the execution to be rather lackluster. In particular, I have questions about the core result (universality); in particular the function class they approximate seems very weak. I also found the presentation very confusing and hard to follow. Finally, the other experiments do not fit the overall narrative and are often underdeveloped.

UPDATE
----------
Lemma 2 alleviates my main concerns. I am greatly pleased by the author response.

---

> ### Author Response · Authors · 2021-11-23
> **Layerwise universality does not imply end-to-end universality in injective flows**
>
> **[Multipart response (part 3/3)]**
>
> *The paper heavily invokes the universality and other properties of its constituent neural networks in the proofs. In particular, most of the proofs are simple applications of universality, follow directly from definitions, and/or rely on canonical tricks in the field (e.g. convolution with a mollifier). Along with the above observation, this lack of depth in the proof gives me the impression that not much real insight is being gained.*
>
> **Response:** Indeed our results are written in terms of constituent networks, and so our networks do depend on their properties. From a theoretical perspective, the manifold-and-density approximation problem may indeed seem like a simple application of known results, but this is far from being the case.
>
> Networks that are distributionally universal, perhaps the most restrictive notion of universality commonly used, are still not `powerful enough' to map one singular measure onto another. Thus one cannot obtain end-to-end universality from layerwise universality. Because of this a weaker notion is needed, which is precisely the embedding gap and the manifold embedding property. Even with universal flow layers there are still problems that cannot be overcome by simply concatenating existing arguments. Because flow maps necessarily learn continuous deformations of the entire space, and not all embeddings can be extended, not all embeddings can be learned. This is not a limit of our analysis, but a topological fact. This leads to the necessity of restricting one's focus to embeddings that can be extended, hence our introduction of the class $\mathcal{I}(\mathbb{R}^n, \mathbb{R}^m)$.
>
> *The paper, as written, is very confusing. This is important as these issues significantly hamper the accessibility of the paper. I would suggest that the authors make a thorough pass through the paper to clear up these issues. I included some of the major ones, but there are certainly others.*
>
> **Response:** We have undertaken a thorough editing effort and cleaned up any of the typos that we found or were pointed out by the reviewers. If you see other things that need fixing we would very much appreciate a pointer.
>
> We have also implemented your suggestion to move what used to be Lemma 1, now called Lemma 7, from the main paper to the appendix. We agree that this improves the overall flow.
>
> *The other results are very ancillary and oftentimes not explored/explained thoroughly. In particular, the last three results seem unrelated (and furthermore do not provide any more real insight). In particular, Section 3.4 (the results on layerwise projection) should be tested empirically. Appendix D.2 (on bayesian uncertainty quantification) is a straightforward application of standard log probability arguments. Finally, Appendix D.3 (the results on black box estimation) seems incomplete. There is no proof for Lemma 8, and I found the last paragraph, which is also one of the few that talks about the current work rather than Rolnick \& Kording (2020), to be very confusing.*
>
> Indeed, the last three properties of the networks feel a bit unrelated. Our aim was to support the claim that these networks have practical merit (and thus the challenge is in analyzing networks with rather constrained architectures rather than arbitrary MLPs), but we have not made it sufficiently clear. We edited the main text and the appendices to clarify these points. We agree that the arguments in Appendix D.2 are more or less standard, hence they are in the appendix, and only mentioned en passant in the main text. We have rewritten the statements and discussions around Appendix D.3 in order to improve clarity. Additionally, we have relabeled what used to be Lemma 8 to a simple remark and instead emphasized exact recoverability of the network, rather than uniqueness of the recovery.

---

> ### Author Response · Authors · 2021-11-23
> **The map you suggest cannot be implemented by the studied neural networks (and thus cannot be considered in proofs)**
>
> **[Multipart response (part 2/3)]**
>
> *For example, a map from $\mathbb{R}^n \to B^n_0(1) \subset \mathbb{R}^m$ is correct topologically but it is not considered in the proofs.*
>
> **Response:** When we consider topological obstructions on the images of the neural networks, our aim is to not only prove the necessary conditions but also to show that there are surprising limitations in that some nice embedded manifolds cannot be image manifolds of the considered neural networks. Our neural networks are compositions of bijective layers and simple dimension increasing maps (e.g., linear maps) which satisfy $\lim_{x\to\infty}|f(x)|=\infty$ (as a pre-image through a diffeomorphism of a compact set) is itself compact and thus bound. We then see that the family of neural networks we consider does not contain maps $f:\mathbb{R}^n \to B^n_0(1) \subset \mathbb{R}^m$. If we misunderstood the nature of your question we would welcome a clarification and we would be happy to answer it.

---

> ### Author Response · Authors · 2021-11-23
> **The class $\mathcal{I}(\mathbb{R}^n, \mathbb{R}^m)$ is optimal + new results for $m > 3n$ using the "clean trick"**
>
> **[Multipart response (part 1/3)]**
>
> We would like to thank the reviewer for taking the time to review our paper. We respectfully disagree with many of the critical points you raise (and we hope to articulate why in the following), but we feel they helped us considerably improve the manuscript.
>
> *The results feel weak. This is best exemplified by the class $\mathcal{I}(\mathbb{R}^n,\mathbb{R}^m)$, which is a very strong assumption to make on our target embedding. This class is very close to the core construction in Brehmer and Cranmer 2020, so a result showing universal approximation does not feel particularly illuminating. In particular, since members of $\mathcal{I}(\mathbb{R}^n,\mathbb{R}^m)$ are $\Phi \circ R$, where $R$ is affine and $\Phi$ is a normalizing flow, do the results not follow simply by approximating both? This feels rather straightforward. Furthermore, while the paper claims that it has to account for just topological obstructions, this class of functions limits for more than just that.*
>
> **Response:** We agree that the class of extendable embeddings $\mathcal{I}(\mathbb{R}^n, \mathbb{R}^m)$ implies a restriction on the class of maps $F$ on whose range $F(\mathbb{R}^n)$ the target measure is supported. Our contribution is to understand in what precise topological and approximation-theoretic sense this can be done, and we feel that this is rather nontrivial.
>
> To illustrate, you suggest that restricting to $\mathcal{I}(\mathbb{R}^n, \mathbb{R}^m)$ is a very strong assumption and state that "while the paper claims that is has to account for just topological obstructions, this class of functions limits for more than just that." Our results, however, show that the class of neural networks that are compositions of bijective layers and simple dimension-increasing layers (such as full-rank linear maps or injective ReLU layers) **exactly coincides** with the class of extendable embeddings $\mathcal{I}(\mathbb{R}^n, \mathbb{R}^m)$. In this sense the class $\mathcal{I}(\mathbb{R}^n, \mathbb{R}^m)$ is optimal—it entails exactly the right amount of restriction.
>
> It is natural to expect that a stronger result is possible—at least we initially thought so. We were surprised to discover that this is actually not the case. It is a mathematical fact that deep neural networks with injective or bijective ReLU (or piecewise linear) layers imply certain topological limitations. In the discussion about the trefoil knot in Figure 2 (and the new example of a knotted 2-dimensional ribbon) we show that these limitations appear in real examples where they prevent the target measure of a neural network to be a certain (rather tame!) class of manifolds. This result in particular means that widely used combinations of flow type networks and linear layers can approximate only a limited class of measures. These surprises are what we believe makes the results particularly interesting. We are aware that this sharpness of $\mathcal{I}(\mathbb{R}^n, \mathbb{R}^m)$ was not clearly articulated in the initial submission; we did our best to properly emphasize it in the update.
>
> We also added a new Lemma 2 (in fact, inspired by the referee's comment) which shows that when output dimension $m$ more than 3 times the input dimension $n$, the limitations are removed and the class of extendable embeddings $\mathcal{I}(\mathbb{R}^n, \mathbb{R}^m)$ coincides with the  class of standard embeddings $\text{emb}(\mathbb{R}^n, \mathbb{R}^m)$. Summarizing, we show that image manifolds of networks consisting of bijective and linear layers are limited, but only when the dimension of the output space is small and when topological reasons  preclude certain target manifolds. This is described in item (ii) of Theorem 1 which applies the proof of item (i) to the composition of layers approximating extendable embeddings.
>
> Proving all these results requires quite a bit of work as well as modern techniques such as the "clean trick" used in the new Lemma 2. These techniques should be of independent interest to the machine learning community, especially as the topological ideas are starting to gain importance. (It is not surprising that occasionally we also need standard devices such as mollifiers; this should not disqualify the rest of the proof or the context in which it is used.) Finally, the studied architectures are indeed not just very close to Brehmer and Cranmer 2020 as the referee points out, but contain it exactly. This is intended, and a strength of our analysis is that it  applies to architectures which were empirically shown to perform well.

---

> > ### Comment · Reviewer_4sjW · 2021-11-24
> > **Commendation.**
> >
> > I commend the authors for their response, and I am glad that I was able to help them make the necessary changes. Lemma 2 is a very strong result, and this ties together the claims of the paper much better. In particular, this is the type of result that better befits the "universality" angle of the manuscript, and I believe that this will be greatly beneficial to the community as the first theorem of this nature.
> >
> > I am currently raising my rating to a 6 under the assumption that most of the author response time was used for this point. Although the PDF can not be updated as of now, lemma 2 should be better worked into the narrative of the paper:
> >
> > * Can the abstract directly tie in the insights from lemma 2? Specifically, I would like some specifics about the result (ie dimensionality and class of functions), as currently the abstract is too high level and covers stuff that is rather ancillary. I would prefer that the abstract mainly talk about this, and not really cover the specific names for the construction and the other appendix material.
> > * In addition to the abstract specifically, I believe that the introduction and other core writing sections should be updated in accordance with the above comments. This in particular means a reframing of the results such as the "Our Contribution" section to include Lemma 2 more precisely. There are many things to change here, so I have definitely missed these points.
> > * Can there be more discussion in the paper about why lower dimensions may fail. In particular, the comment below about the ball embedding (or something like it) would be very helpful for illustration. I understand that the trefoil knot is a good example for why this injective flow trick wouldn't work in general (for pure topology), but an analysis for why neural networks themselves don't work would be rather illuminative.
> > * I would strongly prefer that the appendix material be minimized, as I still do think that they are rather unrelated and, with the above result, serve as more of a distraction. They are rather half-baked, and I would honestly prefer that they be removed entirely. However, I understand that the authors may want to claim primacy on a few of the results, so I am OK with having the layer-wise inversion and bayesian UQ results (as is) in the appendix. However, the black box recovery section is still a massive block of text that reads more as a wistful thought than a concrete method or result.
> >
> > If the authors can examine these, come to an agreement about these points, and commit to updating their manuscript in accordance, I am more than happy to raise my score to an 8. These points should also be made in accordance with the wishes of the other reviewers, as they are rather large/structural changes.

---

> > > ### Author Response · Authors · 2021-11-28
> > > **Demoting**
> > >
> > > **[Multipart response (part 3/3)]**
> > >
> > > Now the second part. From simple linear algebra, we can always find a linear isomorphism $J:\mathbb{R}^m\to \mathbb{R}^m$ that maps $Y$ to any other vector subspace of the same dimension. As $m > 3n$, there is an isomorphism $J$ that maps $Y\subset \mathbb{R}^m$ onto the orthocomplement $Z^\perp$ of a linear space $Z\subset \mathbb{R}^m$ that contains the range of the expansive layer, $R(K)$. Thus we constructed an embedding  $G := J^{-1}\circ Q\circ F:R(K)\subset Z\to Z^\perp$. Let $B=R(K)$. By the 'clean trick' [Madsen 1997, Lemma 7.6], any possibly non-linear embedding $G:B\subset Z\to Z^\perp$ extends to a diffeomorphism $\Psi:\mathbb{R}^m\to \mathbb{R}^m$. For this result it is essential that $G$ maps a subset of a linear subspace of dimension to its orthocomplement and the above assumptions require that the dimension of $Z$ is at least $n$ and the dimension of $Z^\perp$ is larger than $2n$. This is why we need to have $m > 3n$. The clean trick is widely used in algebraic topology to analyze cohomology groups of complements of low-dimensional subsets of $\mathbb{R}^m$.  This extension is the fundamental reason why we can extend the diffeomorphism  $F:R(K)\to \mathcal{M}$ to a diffeomorphism  $\Psi:\mathbb{R}^m\to \mathbb{R}^m$.
> > >
> > > Other ways to sidestep topological issues include allowing $R$ itself to be more topologically complex (that is, allowing $R$ that cannot be represented as $\Phi(\mathbb{R}^n\times \set{0}^{m-n})$ for some embedding $\Phi$; this happens when we choose (R4) as in Example 1), and allowing $R$ or $T$ to be discontinuous or non-injective. We leave this latter idea as an interesting future direction.
> > >
> > > **Comment**: *We could also specifically address your ball example, perhaps as follows:*
> > >
> > > **Response**: Our focus is on maps that are compositions $F=T\circ R$ where $R:\mathbb{R}^n\to \mathbb{R}^m$ is a simple dimension expanding map, for example a full-rank linear map or an injective ReLU-based neural network, and $T:\mathbb{R}^m\to \mathbb{R}^m$ is a diffeomorphism. Instead of diffeomorphisms $T$, one could be interested in other injective maps on $\mathbb{R}^m$, for example, diffeomorphisms $G: \mathbb{R}^m\to B^m_0(1)$, where $B^m_0(1)$ is an unit-radius open ball in $\mathbb{R}^m$ centered at $0$; an example is $G(x)=(1+|x|^2)^{-1/2}x$. However, let us consider
> > >
> > > (i) bijective neural networks, for example the map $T:\mathbb{R}^2 \to  \mathbb{R}^2$, $T(x_1,x_2)=(x_1,x_2+m(x_1))$, where $m(x_2)$ is a multilayer ReLU network, studied in  Dinh et al 2014, or
> > >
> > > (ii) injective ReLU-based neural networks $T:\mathbb{R}^m\to \mathbb{R}^{m'}$,
> > >
> > > In both examples $T$ is a piecewise-affine map and $\mathbb{R}^m$ is a finite union of polygonal convex sets $P_j$, $j=1,2,\dots,J$ in which $T$ is an affine function. Since $T$ is injective in all $P_j$, its derivative is an injective constant matrix in all $P_j$. Thus we see that $\lim_{|x|\to \infty} \frac {|Tx|}{|x|}>0$. Generally, an inverse image of a bounded sequence in a homeomorphism is bounded, so that all homeomorphic maps $T:\mathbb{R}^m\to \mathbb{R}^m$ satisfy $\lim_{|x|\to \infty} {|Tx|}=\infty$. This shows that bijective or injective neural networks have quite different properties than injective maps that map $\mathbb{R}^m$ to bounded subset.
> > >
> > > **Comment**: *I would strongly prefer that the appendix material be minimized, as I still do think that they are rather unrelated and, with the above result, serve as more of a distraction. They are rather half-baked, and I would honestly prefer that they be removed entirely. However, I understand that the authors may want to claim primacy on a few of the results, so I am OK with having the layer-wise inversion and bayesian UQ results (as is) in the appendix. However, the black box recovery section is still a massive block of text that reads more as a wistful thought than a concrete method or result.*
> > >
> > > **Response**: We agree. In light of your comments and the feedback from other reviewers, we see that a narrow focus on universality will make for an overall better and more coherent manuscript. We are debating about how much scope to give to other things in the appendix but in all likelihood we will compress them into a paragraph or two (and mention weight recovery in a sentence). Initially it seemed like a good idea to use them as support but it is now clear that they may be more of a distraction.

---

> > > ### Author Response · Authors · 2021-11-28
> > > **Intuition and explanations about topological obstructions**
> > >
> > > **[Multipart response (part 2/3)]**
> > >
> > > *In addition to the abstract specifically, I believe that the introduction and other core writing sections should be updated in accordance with the above comments. This in particular means a reframing of the results such as the "Our Contribution" section to include Lemma 2 more precisely. There are many things to change here, so I have definitely missed these points.*
> > >
> > > **Response**: We agree. We can commit to such a refocusing of the paper to emphasize universality, especially the new Lemma 2 (that would become a theorem).
> > >
> > > *Can there be more discussion in the paper about why lower dimensions may fail. In particular, the comment below about the ball embedding (or something like it) would be very helpful for illustration. I understand that the trefoil knot is a good example for why this injective flow trick wouldn't work in general (for pure topology), but an analysis for why neural networks themselves don't work would be rather illuminative.*
> > >
> > > **Response**: We can certainly try to add more discussions and intuitive explanations of these technical points from topology. Here is a first suggestion which we will polish further:
> > >
> > > Let $X$ and $Y$ be topological spaces, and let $U \subset X$ and $V \subset Y$. Topological obstructions arise because there exist embeddings from $U$ to $V$ that cannot be extended to an embedding from $X$ to $Y$. As an example, the mapping $f : \set{1,2,3} \to \set{1,2,3}$ where $f(1) = 1, f(2) = 3$, and $f(3) = 2$ is an embedding, but there are no continuous, injective functions from $[1,3] \to \mathbb{R}$  that agree with $f$ on $\set{1,2,3}$. In our problem, a flow layer must be an embedding on all of $\mathbb{R}^m$, not just on the range of the expansive layer. A linear + flow network cannot learn a mapping from $S^1 \subset \mathbb{R}^2$ to the trefoil knot $\Sigma \subset \mathbb{R}^3$ because an embedding $f \colon S^1 \times \set{0} \to \mathbb{R}^3$ such that $f(S^1 \times \set{0}) = \Sigma$ cannot be extended to an embedding on all of $\mathbb{R}^3$.
> > >
> > > The dimension of the ambient space is essential here, since all knots can be opened in a 4-dimensional space but not in a 3-dimensional space. This can be understood by viewing a 4-dimensional space as a space--time  $\mathbb{R}^3\times \mathbb{R}$. Indeed, consider a knotted curve $\gamma$ that exists in the space $\mathbb{R}^3$ at a given time $t_0$. The curve $\gamma$ can  be deformed by moving a piece $\gamma_0\subset \gamma$ to a slightly later time $t_1>t_0$ so that the rest of the curve can freely move through space vacated by $\gamma_0$ at $t_0$. Using such deformations the knot can be deformed to a trivial knot, that is, it can be opened.
> > >
> > > In our case the map $T \circ R$ is a composition of a linear expansive layer $R$ and an invertible neural network $T$, but the above topological issues arise much more generally. Whenever the expansive layer $R$ is an embedding from $\mathbb{R}^n \to \mathbb{R}^m$ that is topologically simple, e.g., a linear map, and the flow $T$ is continuous and injective on $\mathbb{R}^m$, there exist embeddings that $T \circ R$ cannot approximate with arbitrary accuracy.
> > >
> > > Why are such extension always possible for low-dimensional manifolds that lie in a high dimensional space? We give a two-part answer to this question. In the first part, we argue that we can always construct a projection that smoothly embeds $\mathcal{M}$ and all tangent spaces $T_x\mathcal{M}$ into an $m-n$ dimensional linear subspace. In the second part, we argue that we can always find an embedding from this subspace to the range of the expansive layer, and how, by the 'clean trick,' we can always extend this embedding to all of $\mathbb{R}^m$.
> > >
> > > Arguments for why a projection that is an embedding always exists when $m > 3n$ are similar to the proof of Whitney's embedding theorem  [Hirsch 2012, Chapter 1, Theorem 3.4]. Let $F:R(K)\to \mathbb{R}^m$ be  an embedding whose range is a manifold $\mathcal{M} = F(R(K))$ of dimension $n$. Recall that  $m>3n$ and thus $n$ is small compared to $m$.  We can thus always find at least $n$ linearly independent directions $\omega \in \mathbb{S}^{m-1}$ so that the projection operator $P_\omega$ on the subspace $\{\omega\}^\perp$ applied to both $\mathcal{M}$ and  all tangent spaces $T_x\mathcal{M}$ of the manifold $\mathcal{M}$ is injective. By applying the $n$ projections consecutively, we can find a projection $Q:\mathbb{R}^m\to Y$ that embeds  $\mathcal{M}$ smoothly to a submanifold $\mathcal N$ in a linear subspace $Y \subset \mathbb{R}^m$ of dimension $m-n > 2n$.

---

> > > ### Author Response · Authors · 2021-11-28
> > > **Thank you for the very concrete suggestions!**
> > >
> > > **[Multipart response (part 1/3)]**
> > >
> > > Thank you for taking another look at our manuscript, revising your score, and giving us very concrete pointers to further possible improvements. We completely agree with your suggestions for refactoring. It makes sense to better incorporate Lemma 2 in the narrative and place near-exclusive emphasis in the manuscript on universality while demoting UQ and weight recovery. We cannot update the paper right now, but we give concrete examples of possible changes and additions below and we gladly commit to implementing them in the camera-ready version. One high-level change is that given the significance of the new Lemma 2 and the fact that the proof technique may be of independent interest to the deep learning community, we would upgrade it to a theorem.
> > >
> > > *Can the abstract directly tie in the insights from lemma 2? Specifically, I would like some specifics about the result (ie dimensionality and class of functions), as currently the abstract is too high level and covers stuff that is rather ancillary. I would prefer that the abstract mainly talk about this, and not really cover the specific names for the construction and the other appendix material.*
> > >
> > > **Response**: This certainly makes makes sense in the light of new theoretical results and planned removal of UQ and weight recovery from the main text. Here is our first take which we plan to keep working on:
> > >
> > > **Abstract:** We study approximation of probability measures supported on $n$-dimensional manifolds embedded in $\mathbb{R}^m$ by \emph{injective flows}---neural networks composed of invertible flows and injective layers. We show that in general, injective flows between $\mathbb{R}^n$ and $\mathbb{R}^m$ universally approximate measures supported on images of \emph{extendable} embeddings, which are a subset of standard embeddings. When the embedding dimension $m$ is small, topological obstructions may preclude certain manifolds as admissible targets. When the embedding dimension is sufficiently large, $m \geq 3n + 1$, we use an argument from algebraic topology known as the *clean trick* to prove that the topological obstructions vanish and injective flows universally approximate any differentiable embedding. Along the way we show that the studied injective flows admit efficient projections on the range, and that their optimality can be established "in reverse," resolving a conjecture made in Brehmer et Cranmer 2020.

---

> > > ### Author Response · Authors · 2021-11-30
> > > **is this what you had in mind?**
> > >
> > > Dear reviewer 4sjW,
> > >
> > > Before the discussion window closes for good we wanted to quickly ask whether our discussions and update suggestions below are what you had in mind?

---

> > > > ### Comment · Reviewer_4sjW · 2021-12-09
> > > > **Yes**
> > > >
> > > > Yes

---

> > ### Author Response · Authors · 2021-12-09
> > **About your comment "If the authors can examine these, come to an agreement about these points, and commit to updating their manuscript in accordance, I am more than happy to raise my score to an 8"**
> >
> > Dear reviewer 4sjW,
> >
> > At the risk of being annoying, we'd like to ask whether your statement
> >
> > > If the authors can examine these, come to an agreement about these points, and commit to updating their manuscript in accordance, I am more than happy to raise my score to an 8. These points should also be made in accordance with the wishes of the other reviewers, as they are rather large/structural changes.
> >
> > has been addressed by our proposed changes below and commitment to effecting them?
> >
> > Best regards, The authors

---

> > > ### Comment · Reviewer_4sjW · 2021-12-09
> > > **Yes, the proposed changed address what I want.**
> > >
> > > Thank you for providing your updates. They are exactly what I hope for. I am awaiting input from the other reviewers before making the final decision.

---

> > > > ### Author Response · Authors · 2021-12-09
> > > > **reaching out to other reviewers**
> > > >
> > > > Thank you, this is great. Let us try to reach out to the other reviewers.

---

### Official Review · Reviewer_2dK9 · 2021-11-01

**Correctness:** 4
**Technical Novelty And Significance:** 3
**Empirical Novelty And Significance:** 2
**Recommendation:** 6
**Confidence:** 2

**Main Review:**

The paper proposes the concept of an embedding measure reflecting the different topologies, then studies an approximation property of the flow models with distributions with low dimensional support under it. On the positive side, the paper is written in detail, with plenty of examples and discussion.

One question is that it seems to me that it could be written more clearly what kind of problem was being tackled. The motivation for wanting to deal with different topologies is explained, but it seems unclear to me what difficulties were solved as a result. For example, what conclusions can be drawn compared to simply using the Wasserstein distance? Is it possible to construct a mathematical statement whose universal approximability is compromised when existing measures or topology are used? Moreover, it is possible to show that the topology and measure developed in this paper are convincing phenomena, which can be validated by experiments with real data?

**Summary Of The Paper:**

In this paper, the authors studies flow models by a newly developed approximation measure when the data is on a low-dimensional manifold. Specifically, they consider an architecture that alternates a bijective function with the same input and output dimensions, and a function with a larger dimension at the output. There are several successful methods for this, but the globally invertible flow is not well understood. To address this problem, this work shows the approximate properties of the flow architecture. Specifically, they show the approximation accuracy of the models with distributions that have a certain manifold as their support. In doing so, they proposed a new notion of embedding gap to evaluate the manifold embedding. This may allow them to represent cases that cannot be represented by existing topologies. In addition, they defined a value of MEP, which allows us to evaluate the approximation capability under different topologies. This establishes the validity of evaluating the performance for each layer.

**Summary Of The Review:**

Well written. But the contribution could be clearer.

---

> ### Author Response · Authors · 2021-11-23
> **New examples of non-approximability**
>
> We would like to thank you for your positive review and interesting questions which we address in the updated manuscript.
>
> *One question is that it seems to me that it could be written more clearly what kind of problem was being tackled. The motivation for wanting to deal with different topologies is explained, but it seems unclear to me what difficulties were solved as a result. For example, what conclusions can be drawn compared to simply using the Wasserstein distance?*
>
> **Response:** In the revised manuscript we did our best to clearly and directly state the problem being solved. The goal is to build universal approximators, but building them requires overcoming topological problems. We have added to the discussion, and new examples in the Appendix C.2.2 where we build a measure that is non-approximable because of topological difficulties, *even if you use the Wasserstein distance.* In effect, the problem is not with the Wasserstin distance, but rather with the geometry of learning knotted manifolds.
>
> Prompted by your comment we also made additions to Theorem 1 and we added a completely new Lemma 2 which establishes equivalence between $\mathcal{I}(\mathbb{R}^n, \mathbb{R}^m)$ for large enough $m$. We think that these additions make the our results clearer and easier to apply, especially when the dimension of the manifold is much smaller than the ambient dimension.
>
> *Is it possible to construct a mathematical statement whose universal approximability is compromised when existing measures or topology are used?*
>
> **Response:** This is a very useful suggestion that would certainly improve clarity! We added an example of a non-approximable measure at the beginning of Section 3.2.
>
> *Moreover, it is possible to show that the topology and measure developed in this paper are convincing phenomena, which can be validated by experiments with real data?*
>
> Unfortunately we could not produce both bulletproof numerical results and new mathematical results that further clarify our theory. Having to choose between the two we opted to strengthen the main result by proving that the topological obstructions disappear for large enough $m / n$. There exists prior work (namely Brehmer and Cranmer 2020. and Kothari et al. 2021) that shows that the studied architectures produce good-quality output across a range of problems. Let us mention that our paper, similarly to earlier work on universal approximation, is a theory paper. Its goal is not to be of immediate practical use but to show that neural networks used to great effect in practice satisfy certain approximation-theoretic properties. Similar theoretical papers without numerics can be regularly read in machine learning conference  proceedings.

---

> > ### Author Response · Authors · 2021-11-23
> > **A remark about the theoretical nature of our results**
> >
> > We would like to add that while our theory guarantees existence of networks with a particular architecture which approximate any admissible manifold-supported density with arbitrary accuracy, it does not say anything about how to train those networks. Training neural networks with theoretical guarantees is still largely an open question. As a consequence, numerical experiments are uncommon in papers on universal approximation (e.g., Kidger et al. (COLT 2020) “Universal Approximation with Deep Narrow Networks,” S. Park et al. (ICLR 2021) “Minimum Width for Universal Approximation”, R. Gribonval et al. (Constructive Approximation 2021) “Approximation spaces of deep neural networks”, I. Gühring and M. Raslan (2021 Neural Networks) “Approximation rates for neural networks with encodable weights in smoothness spaces”, etc.), since it is challenging to understand whether approximation errors come from optimization or from architectural limitations (or from insufficient sampling).

---

### Official Review · Reviewer_eR1p · 2021-11-02

**Correctness:** 2
**Technical Novelty And Significance:** 3
**Empirical Novelty And Significance:** Not applicable
**Recommendation:** 5
**Confidence:** 3

**Details Of Ethics Concerns:**

N.A.

**Main Review:**

# Post-rebuttal comments (11/29/2021)

I am pleased by the authors that they considered my comments sincerely and revised the manuscript. The updated paper improved the readability. Still, I feel the organization of this paper would have room for improvement. Regarding Lemma 2, I think it improved the applicability of the main theorem.

# Initial Review
【Strength】
- [1] A new measure MEP is proposed for evaluating how much a model can approximate distributions with support in low dimensional manifolds.
- [2] This paper used the concept of MEP to derive the universality of many existing models in a unified manner.

【Weakness】
- [1] I have several questions about the correctness of the statements and proof of the theorems.
- [2] Discussions related to topological obstructions might be hard to follow for those unfamiliar with the (differential) topology.

【Correctness】
- [1] I have several questions about the statements and proof of the theorems.
  - Theorem 1, Cororally 1
    - [1-1] If we read the statement literally, the statement trivially holds by setting $F_i = F$ for all $i$.
    - [1-2] I think this theorem has two definitions of $\mathcal{F}$: the class defined in Lemma 2 and the set of networks of the form Eqn.1 satisfying Point 1--3. Are two $\mathcal{F}$'s different?
    - [1-3] The output dimension of functions in $\mathcal{T}_{\ell}^{n_\ell}$  looks inconsistent. On one hand, according to (1), $\mathcal{T}_{\ell}^{n_\ell} \subset C(\mathbb{R}^{n_\ell}, \mathbb{R}^{n_{\ell}})$ . On the other hand, according to Point 2, $\mathcal{T}_{\ell}^{n_\ell}$ has the $n_{\ell + 1}, n, n_\ell$ MEP, which implies that $\mathcal{T}_{\ell}^{n_\ell} \subset \mathrm{emb}(\mathbb{R}^{n_\ell}, \mathbb{R}^{n_{\ell+1}})$.
    - [1-4] Similar inconsistency happens to $W$ in Cororally 1. The definition of $B_{K, W}(F, E_i)$ implies $W\subset \mathbb{R}^{o}$, while $\mathcal{E}^{n, o} \subset \mathrm{emb}(W, \mathbb{R}^o)$ implies $W\subset \mathbb{R}^{n}$.
    - [1-5] P.17, Proof of Theorem 1: The proof picks $\mu'$ such that $W_2(F_{\#}\mu, E_{\#}\mu') < \epsilon_1 / 2$ . However, I think it is not possible in general. From Lemma 1 Point 5, we only know that $\inf_{\mu'} W_2(F_{\#}\mu, E_{\#}\mu') \leq \epsilon_1$. There is possibility that $\inf_{\mu'} W_2(F_{\#}\mu, E_{\#}\mu') > \epsilon_1 / 2$.
  - Lemma 2
    - [1-6] P.15: According to Definition 3, $W$ must be the closure of an open set of $\mathbb{R}^o$. Therefore, as long as $W$ is non-empty, there exists $w \in W$ and a ball $B$ centered at $w$ such that $B \subset W$. However, $W = K \times \{0\}^{o-n}$ does not satisfy the condition when $K\not = \empty$  and $o > n$.
  - Lemma 7
    - [1-7] P.18, (62): What is the definition of $\nu$?
    - [1-8] P.19: I want to clarify what does Assumption 2 mean. Does it mean there exist calculation procedures for computing the quantities $\log |\det \nabla R_\ell (x_{n_\ell})|$ and $\log |\det \nabla T_\ell (x_{n_{\ell+1}})|$ ? Or does it assume the existence of these quentities (e.g., differentiability of $R_\ell$ and $T_\ell$ and $\det \nabla R_\ell >0$) ?

【Technical Novelty And Significance】
- [1] This paper defined a new measure for measuring the ability of a model for approximating a distribution with support in a low-dimensional manifold. In this respect, the paper is novel. In addition, using this measure, this paper showed the distributional universality of several existing models. Therefore, I think this measure is useful, and introducing it is significant.

【Empirical Novelty And Significance】
- [1] This paper is does not have numerical experiements.

【Detailed Comments】
- [1] P.3: The family of functions $\mathcal{R}_l^{n_{\ell-1}, n_\ell}$ from $\mathbb{R}^{n_{\ell-1}}\to \mathbb{R}^{n_{\ell}}$. : I think this sentence is not a complete sentence and needs rewriting.
- [2] P.3 (R2):  ... , and $W$ is a convlution kernel ... : $W$ → $w$
- [3] P.4: We call a function $f$ an embedding and denote if by $f$ ... → denote it by $f$
- [4] P.4, Definition 3: $\|h\|_{L^\infty(X)} =\mathrm{esssup}_{x\in X} \|h\|_2$ → $\|h\|_{L^\infty(X)} =\mathrm{esssup}_{x\in X} \|h(x)\|_2$
- [5] P.4: When I read the paper for the first time, I could not understand why the inequality $\inf_{\mu_0 \mathcal{P}(W)} W_2 (f_{\#}\mu_n , g_{\#}\mu _o) ≤ B_{K,W} (f, g)$ in P.4 is true. If I understand correctly, it is later shown as Lemma 1 Point 5. So, I would suggest referring to the Lemma when it first appears in P.4.
- [6] P.5: I think it is better to write the definition of equivalent.
- [7] P.5: This is because ... consider for example the exotic spheres from MIlnor (1956): I think those who are not familiar with differential topology may find it difficult to understand this sentence. It would be better to write that what the exotic sphere is (i.e., a topological space that is homomorhpic to but not diffeomorhic to the (hyper)shpere).
- [8] P.4, Definition 3, P.6, Definition 4: In the definition of $B_{K, W}$, $W$ is assumed to be the closure of an open set. However, Definition 4 does not assume so. I am wondering why we need this assumption.
- [9] P.7: This paper used both $\subset$ and $\subseteq$. Does $\subset$ mean $\varsubsetneq$? Or are they just notational inconsistency?
- [10] P.7, Lemma 2: I guess the $\sup$-universal approximator is the concept defined in Techima et al., (2020). So, we need the reference to it.
- [11] P.7, Theorem 1: The definition of distributionally universal is not presented.
- [12] P.8, Lemma 5: universalif → universal if (add a space)
- [13] P.8, Lemma 5: there exists a $f\in \mathcal{F}$ → I think we do not need the indefinite article "a" here.
- [14] P.8, (17): $E_i \circ E_i' \circ T_{i\#}\mu$ → $E_i \circ E_i' \circ T_{i\#}\mu'$.
- [15] P.8: The crux of the problem is ... $\mathcal{R}$ layers when $\mathcal{R}$. → Remove the last "when $\mathcal{R}$"
- [16] P.8: a least squares solution → a least-squares solution
- [17] P.9: For (R3) we have the following result ... : I have a question about this sentence. Since the definition of $R$ in Definition 6 and Theorem 2 is different from (R3), results here are not directly related to (R3).
- [18] P.9: Write the $B$ and $D$ in Definition 6 (I guess it is the same as those in (R3)).
- [19] P.13: The architecture described in Eqn. 22 ... which are not studied here.: I am afraid I could not understand this sentence, especialy the part after "applies to ...". Could you reconsider the sentence?
- [20] P.20: The authors shows that this is possible ... → The authors show ...
- [21] P.20: Further, provided that ... is satisfied, than the entire network can be ... → then

**Summary Of The Paper:**

This paper studied the expressive power of models composed of invertible flows and injective embeddings. First, this paper defined the concept of the Embedding Gap as the measure by which a model can approximate an embedding with low-dimensional support. Then, this paper defined the concept of MEP as the ability to approximate the target embedding arbitrarily small in terms of the Embedding Gap. This paper showed that when invertible layers have the MEP properties and the first layer is a distributively universal approximator (and some additional assumptions), the model is a universal approximator in terms of 2-Wasserstein distance (Theorem 1). Finally, this paper gave a method to compute the inverse transformation of an injective layer with a special form of linear transformation and ReLU nonlinearity. (Theorem 2).

**Summary Of The Review:**

I set the Correctness score to 2 as I have several questions about the correctness of the theoretical part of this paper (see【Correctness】Section). If these questions are solved, I will increase my Correctness score. Regarding technical novelty, I think this paper gave novel tools for analyzing the expressive power of flow models. Introducing these tools is significant as they encompass many existing models.

---

> ### Author Response · Authors · 2021-11-23
> **Thank you for an incredibly detailed review!**
>
> We would like to sincerely thank you for an incredibly thorough review and the many suggested improvements. We describe how we have implemented all of your suggestions below.
>
> ## Correctness
>
> - [1] Theorem 1, Corollary 1
>     - [1-1] The statement has been edited for clarity. $F$ is the function to be approximated and $E_i$ are the pairs of layers of the network.
> - [2] This overloading has been fixed.
> - [3] This has been fixed. The MEP has been changed to apply to $\mathcal{E}$, not $\mathcal{T}$.
> - [4] This has been fixed, a new set $W'$ has been introduced that lies in $\mathbb{R}^n$.
> - [5] This has been fixed. The factor of 2 now carries through the calculation.
> - [6] This has been fixed. We have changed Definition 3 to allow for the case when $W\subset \mathbb{R}^o$ is not itself the closure of an open set, but contains a set $U$ that is an open set relative to a subspace topology of (vector) subspace $V$ of dimension $p$, where $n \leq p \leq o$. This assumption is weaker, still shows that $\textrm{emb}(f(K),g(W))$ is non-empty, and now applies to the proof of Lemma 2, where $K$ is the closure of open set $U\in\mathbb{R}^n$, and $W = K \times 0^{o-n}$.
> - [7] A definition of $\nu$ has been added.
> - [8] What is required is differentiability, i.e., the existence of $\nabla R_\ell$ and $\nabla T_\ell$. The statement and notation of Lemma 7 has been changed to make this requirement clearer.
>
>
> ## Detailed coments
>
> - [1 - 4] All fixed.
> - [5] A reference has been added.
> - [6 - 7] All proposed changes have been made.
> - [8] As discussed above, we have generalized the conditions for Definition 3 to apply such that it no longer requires $W$ to be the closure of an open set. In the statement of the MEP we note that we only need the existence of such an $W$.
> - [9] It was a notational inconsistency, and has been fixed.
> - [10] Reference added.
> - [11] A definition has been added.
> - [12 - 16] Fixed.
> - [17 - 18] Theorem 2 and Definition 6 cover a special case of (R3). This has been clarified with further detail.
> - [19] This part has been clarified. The point is that the mentioned paper also uses quadratic flows which, although not studied directly in this paper, can be addressed using our presented theory.
> - [20 - 21] Fixed.

---

### Author Response · Authors · 2021-12-09
**Reviewers eR1p, 2dK9, WKWr—do you agree with Reviewer 4sjW?**

Dear Reviewers eR1p, 2dK9, WKWr,

Reviewer 4sjW suggested they would further improve their score if we committed to implementing several organizational updates **and if the other reviewers agreed with those updates**. It would be fantastic if you could tell us whether this is the case. The main updates may be summarized as:

- Given the importance of the new Lemma 2, build it more organically into the narrative, and in particular feature the result in the abstract and the introduction; give more intuitive explanations about inapproximable knotted manifolds
- Remove or considerably shorten the discussions about uncertainty quantification and weight recovery which are ancillary to the main universality result

We think that those changes make perfect sense. In our response we gave a detailed draft of the changes and additions we will do (since we cannot update the pdf right now). In particular, we proposed a description of intuition behind knots to put in the main text, and we wrote a new abstract. Our responses are here: [Part 1](https://openreview.net/forum?id=HUeyM2qVey2&noteId=E9rFpB9eN3X), [Part 2](https://openreview.net/forum?id=HUeyM2qVey2&noteId=hkcteH0hHpn), [Part 3](https://openreview.net/forum?id=HUeyM2qVey2&noteId=Q5K8OxH9_oE).

Again, it would be fantastic if you could tell us whether you agree with those updates.

Sincerely,

The authors

---

### Decision · Program_Chairs · 2022-01-20

**Decision:**

Reject

**Comment:**

First this is the seed for a  very good paper on approximating manifolds and densities using injective flows.

Reviewers have done an admirable effort reviewing the paper giving detailed reviews and suggestions to improve the theory and  corrections  that resulted in an improvement of the paper during the rebuttal/ revision phase.

Unfortunately the paper still needs major rewriting and organization to be accessible by other readers, and should undergo another round of review in its last polished version to further vet the correctness of some of its claims as explained below .

The paper was discussed at length among reviewers and the AC and here are the suggestions to improve the paper.

* Implementing Reviewer 4sjW suggestion w.r.t  to the narrative and adding explanations to improve the readability and accessibility  of the paper.

* Another concerns were raised by reviewer eR1p  in the discussion  regarding the correctness of Theorem 1 and Corollary 1. " The proof of Corollary 1 is so rough that I could not confirm its correctness. For example, the functions $r$ and  $f$ are undefined."  Please revisit the proof of this Corollary. Theorem 1 builds on Lemma 7 point 5.  In point 5 of Lemma 7 :"The embedding  $r$ depends on $\epsilon$ , hence so is the measure $\mu$.Therefor the statement  $W_2(g \mu',f \mu)< B_{K,W}(f,g) + \epsilon$ for all $\epsilon$, does not imply that  $W_2(g \mu',f \mu)< B_{K,W}(f,g)$. One solution can be by  building a sequence of measures that would converge to that measure and see if the argument goes through.

 We encourage the authors to implement all the feedback  and suggestions of the reviewers and to submit this interesting work to an upcoming venue.